

# Decline of Etesian winds after large volcanic eruptions in the last millennium

Stergios Misios[1,2], Ioannis Logothetis[3], Mads F. Knudsen[2], Christoffer Karoff[2], Vassilis Amiridis[1], Kleareti Tourpali[3]

[1] Institute for Astronomy, Astrophysics, Space Applications, and Remote Sensing, National Observatory of Athens, Athens, Greece

[2] Department of Geoscience, Aarhus University, Aarhus, Denmark

[3] Aristotle University of Thessaloniki, Thessaloniki, Greece

*Correspondence to*: Stergios Misios (smisios@noa.gr)

## Abstract

Etesian winds represent one of the most stable summer circulation regimes in the Eastern Mediterranean. The Indian Summer Monsoon (ISM) variability and tropical/extra-tropical teleconnections are influencing the Eastern Mediterranean, given that a stronger ISM is often associated with more intense and persistent Etesian winds. The response of Etesian winds to external

forcing on interannual and longer time scales, however, is not well understood. Here, for the first time, we investigate responses of Etesian winds to large volcanic eruptions by analysing a blend of model simulations covering the last millennium and reanalysis data over the 20[th] century. We provide model evidence for significant volcanic signatures, manifested as a robust reduction of the average wind speed in late summer months and the total number of days with Etesian winds. These signatures are attributed to the weakening of the ISM in the post-eruption summer, which reduces large scale subsidence in the Eastern

Mediterranean, weakens the Anatolian low, and finally reduces the intensity and persistence of the Etesian winds. We find a stronger sensitivity of Etesian winds to Northern Hemisphere volcanoes, particularly before the 20[th] century, while for the latest large eruption of Pinatubo, modelled and observed responses are insignificant. Our results could be applied to improve seasonal prediction of wind circulation in the Eastern Mediterranean in the post-eruption summers.

## 1 Introduction

Etesian winds are one of the most stable manifestations of the monsoonal activity in the Eastern Mediterranean (EMed), established by excessive heating in the summer months that builds a steep surface pressure gradient between the Balkans high and the Anatolian low (Carapiperis, 1951; Tyrlis and Lelieveld, 2013). The topography in the EMed channels surface winds to accelerate over the Aegean Sea with an almost northerly direction at the central sector, turning to North-West further south (Dafka et al., 2015; Tyrlis and Lelieveld, 2013). Etesian winds display a pronounced seasonal variation, with peak intensity

and persistence in July and August, as they are  synchronized with the summer monsoon (Dafka et al., 2015). This synoptic system is frequently viewed as the westernmost extension of the Persian trough (e.g. Bollasina and Nigam, 2011) and is thought





to be amplified by a large-scale subsidence established in summer months under the influence of the Indian and Asian summer monsoons (Rodwell and Hoskins, 2001; Tyrlis et al., 2013). The persistent warming by subsidence is compensated by the advection of cool air masses to the Aegean Sea and Levant by the Etesian winds (Ziv et al., 2004), thus regulating summertime

conditions and affecting several socio-economic sectors (Dafka et al., 2018).

Past studies have assessed the frequency and occurrence of Etesian winds on different time scales and forcings. On sub-seasonal scale, increased atmospheric blocking activity is shown to decrease the frequency of Etesian winds, as manifested in the summer of 2014 (Tyrlis et al., 2015). Tropical and extra-tropical teleconnections have also been proposed as an important component on interannual variability. Specifically, the Indian Summer Monsoon (ISM) is thought to influence Etesian winds

by emanating westward-propagating Rossby waves that strengthen subsidence in the EMed (Rodwell and Hoskins, 2001; Ziv et al., 2004; Tyrlis et al., 2013). A stronger ISM, therefore, should favour adiabatic heating in EMed and stronger Etesian wind speeds and vice versa. Logothetis et al. (2022) analysed wind anomalies in the EMed and the strength of the ISM over the period 1900-2000 and found, both in reanalysis datasets and climate model simulations, that intense monsoon activity is associated with increased meridional wind speed in the Aegean Sea. This teleconnection is more prominent in extreme

monsoon years often associated with the El Niño Southern Oscillation (ENSO) (Singh et al., 2020; Kumar et al., 1999). However, the ISM/EMed teleconnection does not systematically hold through the 20[th] century and opposite correlations have been found in the first half of the 20[th] century, suggesting that other mechanisms might be important (Gomez-Delgado et al., 2019). Increased frequency of Etesian winds has been associated with the high pressure system over the Balkans (Metaxas and Bartzokas, 1994; Poupkou et al., 2011) and Chronis et al. (2011) linked the interannual variability of Etesian winds to the

summer-time North Atlantic Oscillation. Climate model simulations under anthropogenic greenhouse gas forcing project a growing number of Etesian days on decadal and longer time scales attributed to the ISM-EMed teleconnection (Dafka et al., 2019; Anagnostopoulou et al., 2013; Ezber, 2019) because monsoon intensity increases in a warmer climate (Sharmila et al., 2015; Kitoh et al., 2013). Such positive trends in model simulations, however, are not supported by observations of Etesian winds over the past decades (Poupkou et al., 2011; Rizou et al., 2018). The response of Etesian winds to external forcing,

therefore, is not well understood.

To improve our understanding of the circulation variability in EMed, a region that has been characterized as a "hot spot" of anthropogenic climate change (Lelieveld et al., 2012), we here investigate possible influences of volcanic forcing on Etesian winds. Explosive volcanic eruptions inject sulphur-containing gases in the lower stratosphere, where they are oxidized to long-lived sulphates that can be globally dispersed within a few weeks (Robock, 2000). These aerosols scatter incoming radiation

and absorb long-wave radiation, hereby altering the global energy budget, which leads to a cooling of the surface (Zanchettin et al., 2016; Timmreck, 2012) and changes in large-scale ocean circulation (Knudsen et al., 2014; Pausata et al., 2015). Several studies have pointed to hydroclimatic responses to volcanic eruptions, characterized by a reduced precipitation in summer monsoon regions (Iles and Hegerl, 2014; Winter et al., 2015; Zuo et al., 2019; Oman et al., 2006; Trenberth and Dai, 2007; Tejedor et al., 2021). The hydroclimatic response is found sensitive to the latitude of the forcing, as tropical and Northern

Hemisphere (NH) eruptions tend to suppress the summer monsoon, opposite to Southern Hemisphere (SH) eruptions (Liu et





al., 2016; Stevenson et al., 2016). A weakened NH monsoon circulation has been identified in the CMIP5 models (Paik et al., 2020), which might be linked to an increased tendency for El Niño warm conditions in the first post-eruption year (Khodri et al., 2017), but the relative amplitude of forced responses compared to natural variability is debated (Dee et al., 2020; Emile-Geay et al., 2008).

In this study, we provide model evidence for a significant reduction of Etesian winds in response to volcanic eruptions over the last millennium, with stronger sensitivity to NH volcanoes. This response is physically explained by the ISM/EMed teleconnection in the summer months. Specifically, strong volcanic eruptions weaken the ISM circulation, reduce large-scale subsidence in the EMed, weaken the Anatolian pressure low and ultimately diminishes the number of summer days with Etesian winds. Aspects of the simulated responses can be found in observations, although of low statistical significance partly

because there are not enough strong volcanic events over the 20[th] century. We conclude by discussing the implications of our results for near-term prediction efforts and how they could be used to better understand Etesian wind changes in a warming climate.

## 2. Methodology

### 2.1 Datasets

We use daily 10m winds, sea-level pressure (SLP), surface temperature and omega velocity from the Last Millennium Ensemble runs carried out with the CESM model (CESM-LME, hereafter) (Otto-Bliesner et al., 2015), which specifies the ice-core–based reconstruction of volcanic forcing by Gao et al. (2008). We present results from a) an ensemble of 12 simulations that consider all known historical forcings and b) a twin ensemble of 5 members considering volcanic forcing only. Those two sets of simulations are merged into a single ensemble (CESM-LME, hereafter) of 17 members spanning over the

850-2005 period and the results are based on this merged dataset unless otherwise specified. From 850 to 1850 we analyze signatures of all strong tropical and high-latitude eruptions according to the classification of Stevenson et al. (2016), while over the 1851-2005 period we additionally consider Krakatau (1883), Santa Maria (1903), Novarupta (1912), Mount Agung (1963), El Chichon (1982) and Pinatubo (1991) (all eruptions are listed in Table 1). However, we find that CESM-LME simulates significant responses for the pre-20[th] century eruptions only, presumably because the historical volcanic forcing was

much larger than the more recent events. For this reason, over the 850-1850 period we discuss results primarily for the strongest tropical eruptions of Samalas (1258), Tambora (1815), and the high-latitude eruption of Laki (1783)[1], while over the 20[th] century we demonstrate responses to Pinatubo only; a selection based on the magnitude of the volcanic forcing in the model. However, similar signatures are found for most of the selected eruptions prior to 1900 as shown in the supplementary figures. The Laki eruption is additionally selected as an example to demonstrate an amplified sensitivity of Etesian wind response to

NH eruptions, and this is further elaborated by comparing differences for all SH, tropical and NH pre-20[th] century eruptions.

---

1 In the CESM-LME, the Laki volcano erroneously erupts in 1762. True date is 1783.





CESM-LME signatures for Pinatubo are evaluated against version 3 of the NOAA-CIRES-DOE 20[th] Century reanalysis Project (20CR, hereafter), which reconstructs past climate over the 1836-2015 period by assimilating into a global climate model historical observations of synoptic surface pressure and specifying sea ice and sea-surface temperatures at the surface

boundary (Slivinski et al., 2019). Dafka et al. (2015) reported an overall consistency in the representation of Etesian winds in different reanalyses compared with station wind observations and similarly our results do not critically depend on the choice of the reanalysis dataset. This is confirmed by additionally considering the updated L-days dataset of Carapiperis (1951), which is an independent observation-based index for the number of Etesian days from 1892 to 2006 that describes Etesian outbreaks when the northerly wind in Athens exceeds the local wind breeze.

**2.1 Definition of Etesian winds**

We calculate daily wind speed (WSP) and wind direction (WDIR) during the late summer season (July and August, JA hereafter), when WSP maximizes, at a fixed grid point (37.5° N, 25.0° E) in the central Aegean Sea, so as to select the typical season and location of the strongest mean wind speed (Tyrlis and Lelieveld, 2013). Following the methodology of Logothetis et al. (2019), a day with Etesian winds occurs when the following criteria are satisfied simultaneously: a) WDIR is between

NE (45°) to NW (315°), b) daily WSP exceeds its long-term median, and c) criteria a) and b) are fulfilled for at least two consecutive days. The latter criterion filters out intermittent disturbances unrelated to the semi-persistent synoptic system of Etesians winds (Hochman et al., 2019). Finally, we calculate the number of Etesian days (NED) per year as the sum of all Etesian days in the JA season. Our definition of the NED index is similar to the "Northerly wind index" of Gomez-Delgado et al. (2019) based on ship-log observations with the addition that our method identifies days with moderate to strong wind speeds

only, which better correspond with the historical L-days index (Poupkou et al., 2011). This is validated by the significant correlation between the NED from the 20CR dataset and the L-index over the common 1892-2006 period (r=0.59, p <0.01 based on a two-tailed t-test). Other methodologies for calculating days with Etesian winds (e.g. surface pressure gradients) products give consistent results (e.g. Dafka et al., 2015).

Despite differences in the period considered and the horizontal resolution that could impact on the representation of the wind

speed climatology, a comparison of the probability density functions of the northerly WSP in the central Aegean Sea finds comparable mean and higher moment statistics, with median values applied to the classification methodology of Etesian winds (second criterion) of 6.8 and 7.1 m/s, respectively (Sup. Figure 1). The NED over the last millennium in one arbitrarily selected CESM-LME run varies from 3 to 50 days, with a median of 21 days and similar statistics are found in the other runs. NED in the 20CR dataset ranges from 9 to 48 days and the L-days index demonstrate a minimum of 3 and maximum of 42 days.

However, the PDF of NED in the observations is skewed to higher NED values compared to CESM-LME (Sup. Figure 1b), likely related to the horizontal resolution of the CESM model.

**3. Results**



### 3.1 Reduction in the number of summer days with Etesian winds

We firstly analyse the ensemble mean time series of the Northerly WSP (criterion a, see section 2.2) and NED, respectively,
over the last millennium, as simulated in the all-forcing (black line) and volcanic-only forcing (cyan line) CESM-LME
ensembles (Figures 1a and b). Periods of muted interannual variability are punctuated by periods of enhanced activity, with
some evidence for positive trends over the last century, suggestive of an increasing number of Etesian days and WSP in the
all-forcing simulation. This is consistent with independent model simulations of future greenhouse gas emissions resulting in
higher WSPs and NEDs (Dafka et al., 2019; Anagnostopoulou et al., 2013). The most notable deviations of WSP and NED,
however, are found in volcanically active years. Major volcanic eruptions, as noted by increased outgoing NH clear sky SW
radiation (grey line in Figure 1) that is used as a proxy of stratospheric aerosol loading, frequently reduce the northerly WSP
and consequently the NED up to two years after the eruption. For Samalas, the largest eruption in the last millennium, CESM-
LME simulates negative WSP anomalies exceeding -1.3 m/s in the two post-eruption years, while the ensemble mean NED
hardly exceeds 10 days in the summer of 1259. The second strongest WSP and NED reduction is found for the NH Laki
eruption, even surpassing changes associated with Kuwae, the second strongest eruption in the last millennium and all other
tropical eruptions (e.g. Tambora).

Figure 1c zooms over the common 1836-2006 period to compare all-forcing CESM-LME with the 20CR and L-days. We
caution about the first years in 20CR given the scarcity and quality of observations, but this early period is nevertheless
irrelevant for our analysis because of the lack of any large eruptions. 20CR does not record any significant NED reduction
after Krakatau, broadly consistent with CESM-LME. On the other hand, the Pinatubo eruption reduces NED in the summer of
1992 (and wind speed, not shown) as evidenced in both the 20CR and L-days datasets. It is interesting to note that the absolute
minimum NED anomaly is found in the summer of 1913, with only 8 and 3 days respectively in 20CR and L-days, which
could be associated with the Novarupta-Katmai eruption in Alaska (Wes Hildret and Fierstein, 2012). However, this
postulation is not supported by the CESM-LME runs, possibly because of the unrealistically weak forcing imposed in the
150   model. Such an underestimation of the NH volcanic forcing is commonly found in many forcing datasets used in model
intercomparison activities (Toohey et al., 2019). A comparison of the NH clear sky TOA outgoing SW radiation finds about
x5 stronger anomalies in the 20CR compared with CESM-LME, further supporting this possibility (not shown).

The previous discussion highlighted a tendency for reduced NED in volcanically active periods. This is further substantiated
with a superposed epoch analysis of NED in the 5 post-eruption years (years 1 to 5) compared to the pre-eruption 5-year
155   average (years -5 to -1, see Figure 2). To facilitate the comparison, anomalies are given in percentages. In addition, Sup. Figure
2 presents modelled responses for all selected eruptions from 850 to 2000. After Samalas, the NED declines in all individual
runs (17 thin grey lines) with maximum anomalies up to -90% peaking at year +1. The large number of realizations in CESM-
LME facilitates the detection of volcanic signatures versus natural variability (Stevenson et al., 2016) and we detect a
significant response exceeding 2 standard deviations of the previous five years. The absolute minimum NED in the summer
of 1259 is 3 days, found in two runs, essentially describing a summer without Etesian winds. Similar summers are also





simulated after Tambora, but with higher intra-ensemble spread regarding the timing of the peak reduction, given that NED in individual runs minimize either in year-0 or year +1. As for strong explosive eruptions, the effusive Laki eruption causes a significant NED decline, with the strongest ensemble mean reduction of -60% found in year-0. In individual model runs, NED anomalies are as large as those of Samalas, with magnitudes up to -80%. Because of the high latitude and the relative low

altitude of the eruption, the lifetime of the Laki influence is relative short; NED returns to the pre-eruption conditions the following year. Yet, Pinatubo does not significantly impact the ensemble mean NED (Figure 2d), despite the fact that some runs show strong reductions with amplitudes exceeding -60%. The peak response of -20% at year-0 is not significant and in addition is suspiciously early as it peaks just two months after the eruption (June 1991). In observations, NED drops by 20-40% in the summer of 1992 both in 20CR and L-days, but the signal is neither significant based on a t-test, nor exceptional as

NED also drops by about the same magnitude at year -4. Hence, we conclude that the observed NED reduction in the summer of 1992 is not significant and might not be related to the volcanic forcing, which is consistent with the model results. This is further discussed in the following Section.

As the historical volcanic forcing is larger than recent events (e.g. see gray line in Figure 2), it is not surprising that the CESM simulates a robust ensemble mean NED decline for volcanoes prior to 1900 only, while signatures are inconsistent over the

20[th] century. In addition, not all strong eruptions impact Etesian winds in the same way because of interhemispheric differences in the forcing. This is demonstrated in Figure 3, which shows NED anomalies separately for all SH, tropical, and NH eruptions from 850 to 1900 (bold eruptions in Table 1). For the SH eruptions, only Kuwae causes a significant NED anomaly but nevertheless of considerably weaker magnitude compared to other volcanoes. Tropical eruptions typically reduce NED in year +1, as also demonstrated for Samalas and Pinatubo (Figure 2 and Sup. Figure 2). NED also reduces after all NH eruptions,

with a multi-eruption average anomaly of -30%, larger than the mean response on tropical eruptions. This amplified sensitivity to NH eruptions is consistent with studies that show disproportionally stronger climate forcing between NH high-latitude and tropical eruptions of equal magnitude (Toohey et al., 2019; Liu et al., 2016).

**3.1 Waning Indian Summer Monsoon reduces summer days with Etesian winds**

CESM-LME simulates a decline of NED of considerable magnitude in the post-eruption years, which additionally is found

sensitive to the hemisphere of the volcanic forcing. To understand these key findings, we need to investigate the large-scale circulation changes in relation to the ISM. As a first step, we analyze large-scale surface temperature, SLP, and wind anomalies in the post-eruption summers using monthly mean data.

Previous analyses of the CESM-LME simulations have identified the coldest annual NH temperatures after major volcanic eruptions, with magnitudes that generally are stronger than in the reconstructions, possibly related to uncertainties in the

specified volcanic forcing (Otto-Bliesner et al., 2015). Likewise, the strongest summer (JA) cooling in Southern Europe/Northern Africa is simulated after Samalas, with anomalies exceeding -3.5 K (not shown). An extensive cooling in the EMed, Balkans and Levantine is also simulated after the Laki and Pinatubo eruptions, partly explained by the reduction in the incoming radiation by the volcanic aerosols in the stratosphere. This direct radiative cooling is superimposed on dynamical



signatures associated with changes in the large-scale circulation and regional land/sea contrasts. To highlight these dynamical

signatures on EMed, Figure 4 shows surface temperature anomalies (color shading) in the first post eruption summer (JA) after zonal mean temperatures (e.g. about -2 K in the 30-40°N) are subtracted. This approach isolates an amplified cooling over the EMed and the Arabic peninsula. This pattern is similar to all eruptions (see also Sup. Figure 3) with the strongest magnitude of -1.5 K in EMed found for Samalas. The sharp land-sea temperature gradient is related to the heat capacity of water that dumps a response to an intermittent forcing. The cooling is associated with an increased SLP up to 3.5 hPa in the case of

Samalas and Laki. This indicates a shallower Anatolian low in summer, which in turn weakens the SLP pressure gradients over the Aegean Sea and reduce wind speeds as evidenced with the southerly anomalies of about 1 m/s (Figure 4). Hence, fewer days with Etesian winds are expected under such surface conditions as indicated in Figure 2.

The positive SLP anomalies extend throughout the Middle East and the Arabian Peninsula. The north-east wind anomalies in the Arabian Sea and the Bay of Bengal oppose the prevailing winds in the summer season affecting the ISM region; this might

be related to weakening of the ISM that can explains the anomalous heating over India by a reduction in the cloud amount and increasing shortwave heating (Dogar and Sato, 2019). This pattern is robust and is simulated for all strong NH and tropical eruptions prior to 1900 (Sup. Figure 3). It is interesting to note that the forced ensemble-mean surface warming in India is the strongest for Laki, exceeding +2.5 K. Likewise, there is also some evidence for north-west wind anomalies in the Arabian Sea after Pinatubo in the summer of 1992, but the CESM-LME surface cooling in the EMed is considerably weaker. SLP anomalies

in the Anatolian low are trivial after Pinatubo, which is consistent with the insignificant NED reduction shown in Figure 2d. This pattern of forced response in the temperature and surface winds, which has also been simulated with other models (Dogar and Sato, 2019), suggests a possible connection to the ISM. This possibility is investigated by analyzing the omega velocity fields at 200 hPa (Figure 5 and Sup. Figure 4 for all eruptions). The climatology of omega velocity in JA is characterized by ascending motions over the Bay of Bengal, India and Nepal (contours in Figure 5), linked to the monsoonal activity, while

subsidence prevails over the region of EMed (Rizou et al., 2018; Logothetis et al., 2019). These two opposite vertical motions are thought to form a closed circulation system during the summer months, linking the Indian and South Asian summer monsoons to the circulation in the EMed (Rodwell and Hoskins, 2001; Tyrlis et al., 2013). CESM-LME simulates positive anomalies in the region of the Indian monsoon after the volcanic eruptions of Samalas, Pinatubo and Laki, indicating a significant reduction of the upward motion and a waning monsoon activity, which can also be inferred by the reduced

precipitation (not shown). The anomalous descending winds in the ISM region is paired with negative anomalies over the EMed, indicating a reduced subsidence in the post eruption years. A comparable pattern, although of weaker magnitude, is obtained at 500 hPa (not shown). Anomalies are significant at $p < 0.05$ based on a two-tailed t-test and the strongest changes exceeding 0.04 Pa/s (-0.02 Pa/s) in the ascending (descending) branch are found for the Samalas eruption as perhaps expected from the strongest decline in the Etesian winds.

Previous studies detected a substantial decrease in precipitation over land after the Pinatubo eruption, associated with a reduction of the ISM and positive surface temperature anomalies over India (Trenberth and Dai, 2007). This is consistent with the surface temperature and wind anomalies found in 20CR (Figure 4e) and broadly supports the pattern of warming in India



and cooling in the EMed found in CESM-LME. Yet, the surface temperature response after Pinatubo is considerably weaker in CESM-LME compared to 20CR. The reduced ISM in the summer of 1992 causes positive anomalies in the ascending region but the observed signature in the descending branch in EMed in negligible and insignificant (Figures 5e). This is consistent with the weak SLP anomalies shown in Figure 4e, associated with insignificant anomalies in observed NED.

Despite the inconsistent signatures after Pinatubo, CESM-LME simulates similar patterns of omega velocity anomalies for all tropical and NH eruptions prior to 1900 (Sup. Figure 4). Moreover, we identify an almost linear relationship between changes in ISM strength and NED anomalies. In Figure 6, changes in the ISM strength are approximated by the omega velocity anomalies at 200 hPa averaged in the black boxes of Figure 5, which delineate the region of the strongest mean ascending motion. This definition has been applied in previous studies, albeit averaging over slightly different regions of the ISM (Logothetis et al., 2022). The omega velocity anomalies display a negative correlation to the NED anomalies (r=-0.8, p <0.01 according to a two tailed t-test) in the post-eruption year, and a linear regression provides a negative slope of 3-2.3 days per 0.01 Pa/s increase of anomalous omega velocity, significant with p <0.01 based on a two-tailed t-test. If we additionally consider the SH eruptions, the linear regression shows a steeper slope of -3.5 days per 0.01 Pa/s, attributed mainly to the outlier of 1341 eruption which shows positive NED and omega anomalies (Sup Figures 2,3).

## 4. Discussion and Summary

Large eruptions make ideal test cases for evaluating the climate response to external forcing and improve our understanding of the mechanisms mediating global signatures to regional scales (Soden et al., 2002). Using CESM-LEM last millennium ensemble, we investigate, for the first time, volcanic influences on Etesian winds in post-eruption summers. The ensemble mean response is characterized by anomalously colder summers in the Mediterranean after all last-millennium eruptions, an effect that is the strongest in the EMed (Sup. Figure 3) because of reduced adiabatic heating (Sup. Figure 4). Reconstructions of summer temperatures indicate that several cold spells in the EMed often coincide with volcanic eruptions (Klippel et al., 2019; Klesse et al., 2015), but the strongest cooling in the last millennium is not associated with the strongest eruptions (e.g. Samalas), which contrasts with the CESM-LME results. However, this should be expected given that the ensemble averaging in CESM-LME suppress internal variability. Volcanic eruptions are found to impact SLPs primarily over the Anatolian Low, whereas changes associated with the Balkan SLP high are trivial. This weakens SLP gradients over the Aegean Sea, reduce winds speeds, and diminish the number of summer days with Etesian winds. According to the CESM-LEM, the year of 1259 could have been a summer without Etesian winds, provided the internal variability had been negligible.

Tropical and NH eruptions suppress convection over the warm pool, reduce ISM precipitation, and weaken monsoonal circulation as indicated by the weakened ascending motions reflected in the omega velocity anomalies (Figure 5). A suppressed ISM in the post-eruption summer is associated with reduced subsidence in the EMed and higher SLPs along the western margin of the monsoon (Ziv et al., 2004). This explains the SLP anomalies in the Anatolian low and hence the simulated decline of NED after strong volcanic eruptions. The NH eruptions typically lead to the strongest latitudinal temperature gradients, given that they induce negligible cooling the tropics (Stevenson et al., 2016). This leads to an enhanced monsoon suppression




compared to tropical and SH eruptions, which explains the stronger sensitivity of NED. Yet, CESM-LME does not provide evidence that a tropical eruption of Pinatubo-like magnitude can alter NED significantly. This is supported by the observations, which also show insignificant anomalies. On the other hand, a Pinatubo-size NH eruption should cause a considerable reduction in Etesian winds owing to the amplified sensitivity.

Our results are based on simulations with a CESM model that assumes some simplifications regarding aerosol transport in the stratosphere, aerosol distribution, and the seasonality of eruptions (Gao et al., 2008). Eruptions of unknown dates are assumed to begin in April and peak in June-July, a simplification that leads to very similar time evolution of the volcanic forcing (Stevenson et al., 2016). This might lead to an overestimation of the volcanic forcing in the summer months for some the unknown eruptions. CESM also suffers from over-active ENSO variations compared to the observations, which need to be

considered when disentangling direct volcanic effects and ENSO. By averaging over 17 ensemble members, the effects of ENSO on ISM should be considerably alleviated but there is some evidence that volcanic eruptions in the CESM-LEM promote positive? ENSO conditions (Stevenson et al., 2016). This means that, in the model, the monsoon response to the volcanic forcing could have been amplified by an ENSO warming in the post-eruption year. Observations and reconstructions do not still provide undisputable evidence regarding the ENSO response to volcanic forcing (Dee et al., 2020; Khodri et al., 2017).

We conclude that the suppressed monsoon and the ISM/EMed teleconnection is mediating global volcanic signatures to EMed, affecting the synoptic pattern of Etesian winds. These findings could help us separate naturally and anthropogenically forced variations. Model simulations of future global warming indicate a strengthening of the land–ocean temperature contrasts and low-level monsoon circulation, accompanied by enhanced precipitation over the ISM region (Kitoh et al., 2013; Sharmila et al., 2015). According to our results, an intensification of the ISM under increased greenhouse gas forcing might strengthen the

Anatolian low and ultimately increase the NED in future summers. Evidence for intensified Etesian winds has been inferred from simulations of future scenarios (Ezber, 2019; Anagnostopoulou et al., 2013), suggesting nevertheless that additional mechanisms related to changes of the midlatitude westerly flow might also play an important role in strengthening SLP gradients in the Aegean Sea (Dafka et al., 2019).

Prediction of the frequency, intensity, and persistence of Etesian winds in summer months is important for ecosystem services,
wildfire prevention, air quality forecasts, tourism, and economic development. The synoptic system associated with the Etesian winds exhibit high predictability compared to the other weather patterns in the EMed (Hochman et al., 2019). Given the recent progress in seasonal predictions of the ISM after volcanic eruptions (Singh et al., 2020), our results could be used for improved seasonal predictions of wind circulation in summer months in the EMed.




## Author contributions

SM designed the analysis and wrote the manuscript. IL, MFK, CK, VA and KT contributed to the manuscript and provided feedback.

## Data and code availability

Data from the Last Millennium Ensemble Project with the CESM model are available here https://www.cesm.ucar.edu/projects/community-projects/LME/. Data from the 20th Century reanalysis project are available here: https://psl.noaa.gov/data/gridded/data.20thC_ReanV3.html. Code for the analysis is available upon request.

## Competing interests

The authors declare that they have no conflict of interest.

## Acknowledgement

This work is funded by the MSCA action "Climatic impacts of volcaninc ash electrification-ElectricVolcano". SM. M.F.K and C.K aknowledge the Villum Foundation Experiment Programme 'Environmental consequences of solar cosmic rays'. We thank Evangelos Tyrlis (National and Kapodistrian University of Athens) for helpful discussions.

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





Table 1 List of eruption years considered in this study. Eruptions from 850 to 1850 are classified to Southern Hemisphere, tropical and Northern Hemisphere following the methodology of Stevenson et al. (2016). In bold are the SH, tropical and
NH pre-20th eruptions used in Figure 3.

| Classification of Eruption Forcing | Eruptions |
|---|---|
| Southern Hemisphere | **1275, 1341, 1452** |
| Tropical | **1258, 1284, 1809, 1815, 1883**,1963, 1982,1991 |
| Northern Hemisphere | **1176, 1213, 1600, 1641, 1762, 1835**, 1903, 1912 |



Figure 1 Time series of WSP (m/s) and NED (days) over the last millenium (850-2005) in CESM-LME all forcing ensemble (black), CESM-LME volcaninc-only forcing ensemble (cyan), 20CR (red) and L-days (orange). a) Ensemble mean summer



(JA) WSP time series. b) Ensemble mean summer (JA) NED time series. c) as panel b) but for the shorter period 1836-2005 and including 20CR and L-days. Grey lines (relative scaling) show the outgoing clear sky SW radiation in the Northern Hemisphere, which is a proxy of stratospheric aerosol loading.

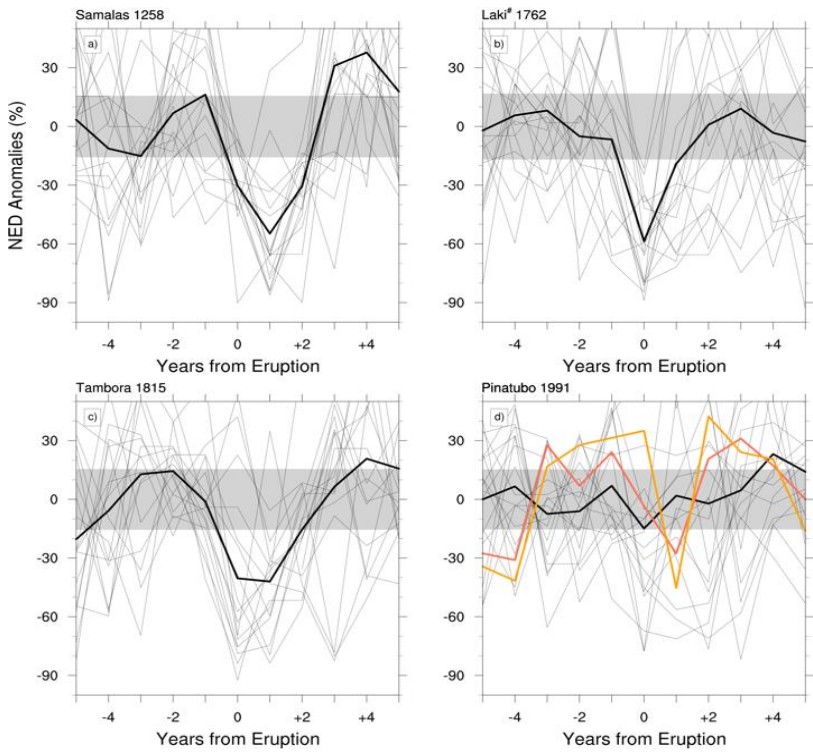

Figure 2 Percentage anomalies of NED for (a) Samalas, (b) Laki, (c) Tambora and (d) Pinatubo eruptions in CESM-LME all
forcing (black), 20CR (red) and L-days (orange), from 5 pre-eruption to 5 post-eruption years. Zero denotes the year of eruption. Thin black lines show twelve individual runs of the CESM-LME all forcing ensemble. Grey shading measures the +-2 standard deviation of pre-eruption (-5 to -1 years) ensemble spread. Note that the Laki eruption is erroneously specified in 1762 in CESM-LME. See Sup. Figure2 for all other eruptions in the last millenium.

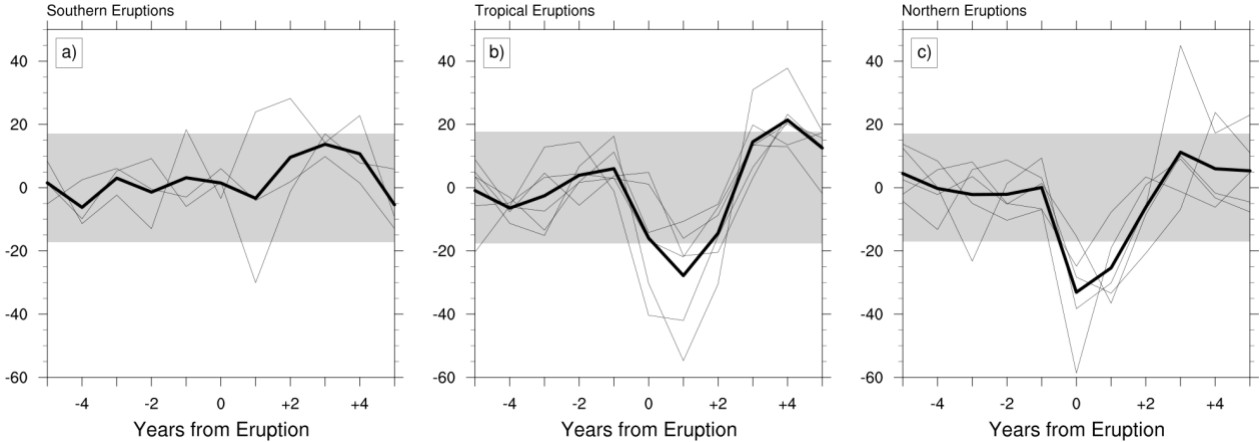





Figure 3 Percentage anomalies of NED for (a) southern hemishere, (b) tropical and (c) Nothern hemisphere volcanic eruptions from 850 to 1900. Zero denotes the year of eruption. Thin black lines show the ensmeble mean CESM-LME response in individual eruptions and the thick black line the multi-eruption mean response. Grey shading measures the +-2 standard deviation of pre-eruption (-5 to -1 years) spread. Southern eruptions: 1275, 1341, 1452, tropical eruptions: 1258, 1284, 1809, 1815, 1883, and Nothern eruptions: 1176, 1213, 1600, 1641, 1762, 1835, see Table 1.

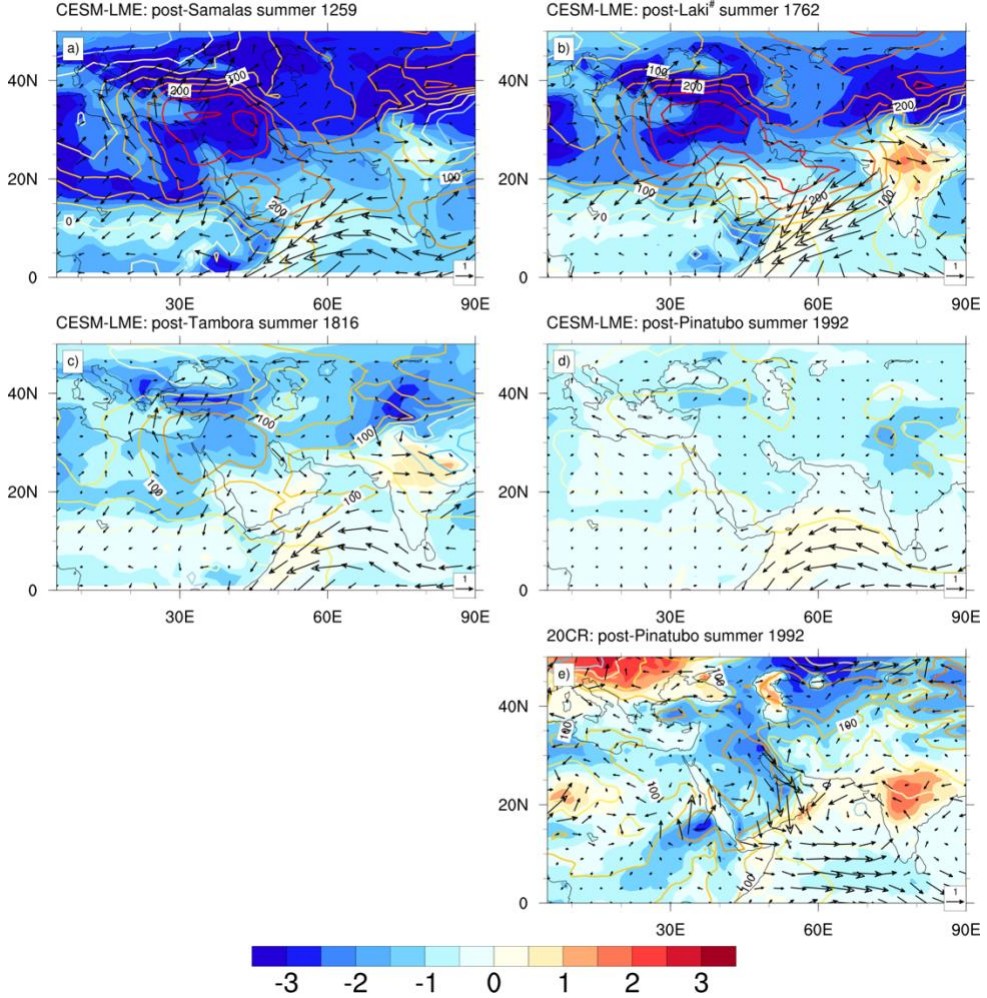


Figure 4 Anomalies of surface temperature (shaded), 10m winds (arrows) and SLP (contours) in the post-eruption summers (after Samalas, Laki, Tambora and Pinatubo) of maximum NED response. Panels a-c) from CESM-LME ensemble and e) from 20CR. Anomalies relative to the average 5 years before the eruption year (see Table 1). No significance test is overlayed. See Sup. Figure 3 for all other selected eruptions in the last millenium.

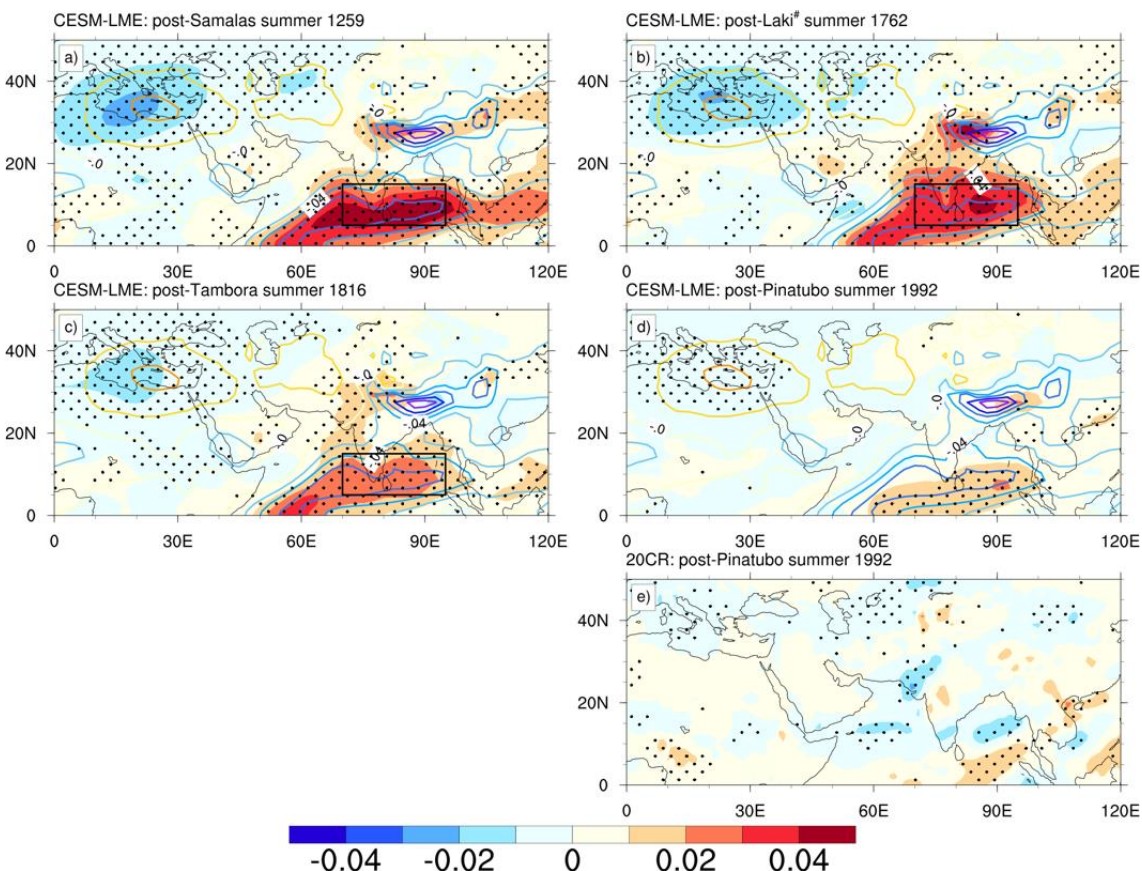


Figure 5 Anomalies of omega velocity (-Pa/s) at 200 hPa in the post-eruption summers (after Samalas, Laki, Tambora and Pinatubo) of maximum NED response. Panels a-c) from CESM-LME ensemble and e) from 20CR. Positive (negative) values indicate reduced ascending(descending) motions. Anomalies relative to the average 5 years before the eruption year (see Table 1). Regions of $p < 0.05$ based on a two-tailed t-test are stippled. See Sup. Figure 4 for all other selected eruptions in the last

millenium.

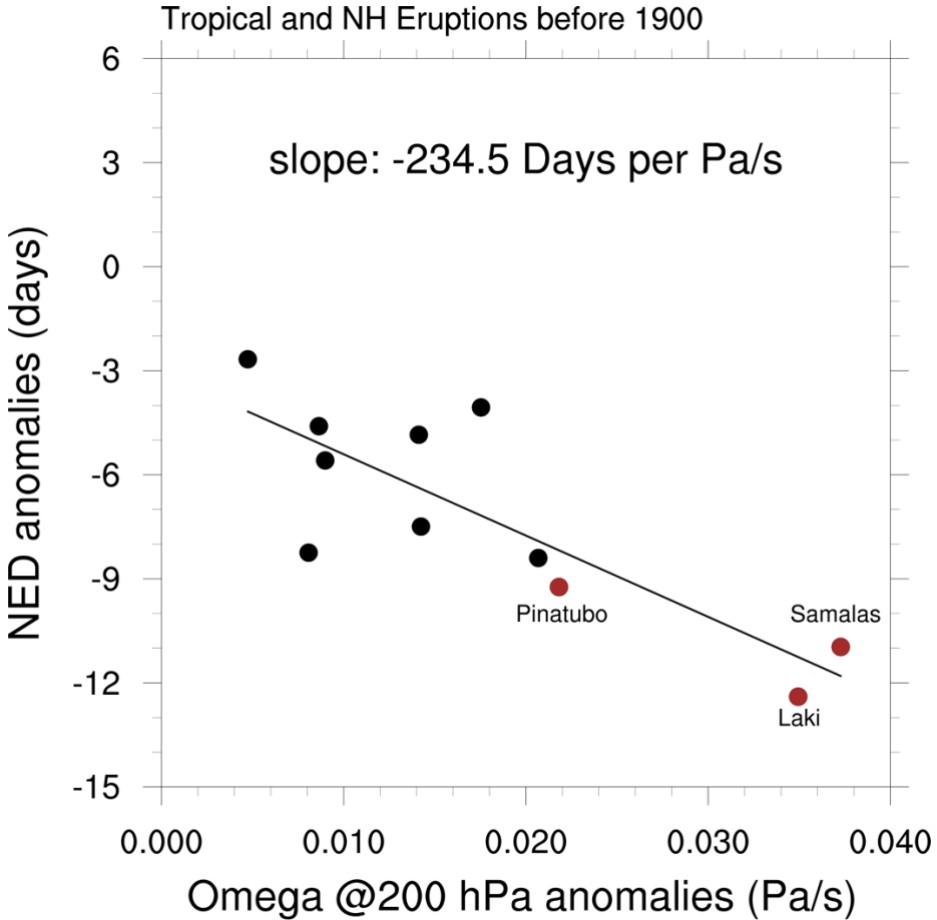

Figure 6 Relation of the JA omega velocity anomalies (-Pa/s) at 200 hPa averaged over the mean ascending region (black boxes in Figure 5) to the NED anomalies in the CESM-LME. Tropical and NH volcanoes (see Figure 3) before 1900 are used only. Brown circles denote Pinatubo, Samalas and Laki. A linear regression calculates a negative slope of about -2.3 days per 475 0.01 Pa/s significant at p <0.01 based on a two tailed t-test.