# Peer review of "Decline of Etesian winds after large volcanic eruptions in the last millennium"

_Weather and Climate Dynamics, 2022_

## Referee Comment (RC1)

Review comments on the submitted article "**Decline of Etesian winds after large volcanic eruptions in the last millennium**" by Misios S, Logothetis I, Knudsen M F, Karoff C, Amiridis V and Tourpali K.

**General comments**

This study uses data from the 20$^{th}$ Century Reconstruction project (20CR) and the output from the Last Millennium Ensemble simulations with the model CESM (CESM-LME) to investigate the impact of large volcanic eruptions on the summer circulation over the eastern Mediterranean (EMed) and, in particular, the persistent northerly flow, known as the Etesians. The authors present evidence that major volcanic eruptions in the Northern Hemisphere are able to weaken the pressure gradient over the EMed, reduce the intensity of the Etesians and their frequency of appearance. It is also shown that the well-known monsoon-desert mechanism weakens during summers after major eruptions. The effect of eruptions appears to be more obvious for eruptions before the 20$^{th}$ century when the volcanic radiative forcing is considered to be stronger. The work bears the necessary scientific merit and I reckon that it can have an important contribution to the topic because it is characterized by the element of novelty. However, the manuscript suffers at the writing and presentation level. Frequently, there is incomplete discussion on the results, the uncertainties involved and the dynamics explaining the aforementioned link. My recommendation is that the manuscript could only be ready for publication by Weather and Climate Dynamics after major revisions, mainly due to the large number of suggested comments. Below I make specific suggestions that aim at improving the manuscript.

**Specific Comments**

**1) Presentation of the coincidence between volcanic eruptions and reductions in the frequency of Etesians**: The presentation of results in Figure 1 should to be improved:

i) Overlaying the cyan and black lines (Figs. 1a and b) that represent CESM-LME all forcing ensemble and CESM-LME volcanic-only forcing makes it difficult to distinguish between the two. Perhaps I miss something but actually, in the text there is very limited discussion on the difference between the two that draws conclusions on the importance of the volcanic forcing alone compared to the all-forcing simulations. If so, does it make sense to show here only one curve? Otherwise, please comment on the difference between the two and the necessity of showing both of them here.

ii) Caption and description of Fig. 1c are problematic. It is not clear what the red and black lines represent. I can only speculate that the black line is the same as the black line (?) in panels 1a,b. But again the vertical range is suppressed and the line appears to very flat, which hinders the effort to draw conclusions on the timing with the volcanic eruptions.

iii) Grey line gives the outgoing clear sky SW radiation in the Northern Hemisphere, but units are not given here except that "scaling is relative". Perhaps units should be also given on the right hand side vertical axis.

iv) L135, L151 and caption of Fig. 1: More on the previous comment. NH, SW and TOA are not explained here. Could you please give more information about the physical basis that relates volcanic eruptions and coincident higher outgoing SW? Does "outgoing clear sky SW radiation" refer to radiation reflected by aerosols in cloud-free regions?

v) I find extremely hard to locate the dates of volcanic eruptions, which correspond to peaks in the grey line (Fig. 1) so as to check the coincidence with the weakening of the Etesians over the EMed. The authors should add several vertical lines that mark all major eruptions mentioned in Table 1. A solution could be to add an initial letter (i.e., P: Pinatubo) that should be also indicated in Table 1.

2) **Details about CESM-LME simulations.**

i) L82-85: Here the authors describe briefly the sets of simulations used but they do not disclose details on how the initialization of the various ensemble members was performed. What is the meaning of "twin ensemble of 5 members"? How was the initialization of these ensemble members made?

ii) Also, it may be helpful that the authors use different acronyms for results obtained with the use of simulations from CESM-LME all forcing, only volcanic forcing simulations and for the cases that all 17 members of CESM-LME are grouped together. For example something like, CESM-LMEaf, SESM-LMEvf and CESM-LMEt, respectively. These acronyms could be used consistently throughout the manuscript to alleviate confusion.

iii) Details of the spatial resolution of the CESM-LME simulations and the 20CR dataset should be also listed. In L125-126, the authors argue that PDF of NED is skewed towards higher values in 20CR when compared to the one corresponding to CESM-LME. But it is not explained how the resolution is expected to affect the PDF. Please clarify.

iv) L157-158 & 270-272: I am not sure that the ensemble size of 17 members is adequate to ensure that volcanic signature will emerge from natural variability. And even more so with respect to the effect of the ENSO on ISM. In recent studies, there is a tendency for the use of sets of at least 100 ensemble members to disentangle signal form natural variability.

v) L89-90 & L92: Is the volcanic forcing applied to the simulations available so as to confirm the hypothesis that the historical forcing was stronger? The terms "historical" and "recent" are frequently used in the manuscript, but sometimes very loosely.  Please clarify.

3) **Detailed discussion on the synchronicity between volcanic eruptions and decline in the frequency of Etesians in Fig. 1c**:

i) L142: As mentioned above, the meaning of the dark and red lines is not clarified. I speculate that the red line represents NED frequency based on 20CR.

ii) L144: I disagree with the statement that both 20CR and CESM-LME do not show any significant decline after Krakatau (1883).  Perhaps I miss something but I see many minima both in black and red lines (especially the red line – 20CR) in the 1890s, 1910s, 1930s, 1970s & 2000s. This is actually confirmed in the next sentence that refers to the eruption of Pinatubo, which is confusing.

iii) L149: Is it possible to check whether the forcing is actually realistic in CESM-LME simulations specifically during this eruption?

iv) L153: This is very true for 20CR and L-days but may be not so accurate for CESM-LME. But again the vertical axis range used to plot the black line in Fig. 1c should be different so as to depict finer variability. Now the line appears rather flat.

v) Surprisingly, there are many other minima in the red line that are not associated with volcanic eruptions. See for example the one a few summers prior to the 1883 Krakatau eruption. While many maxima in the grey line are not associated with minima in NED frequency, such as the case of early 1960s (1964?). This points out the relevance of other dynamical mechanisms (both in the tropics and mid-latitudes) that influence the summer circulation over the EMed.

4) **Discussion of the "epoch analysis" in Fig. 2**:

i) L154-155: Have you calculated the anomalies shown with respect to each ensemble separately?

ii) L156-157: In the text the authors mention that 17 thin grey lines represent the evolution for all 17 ensemble members. But in the caption of Fig. 2, it reads that results are shown for the 12 ensemble members of CESM-LME all forcing. Please clarify.

iii) L157: It could be helpful to draw with different colors the curves representing the anomalies for the 5 CESM-LME members that use volcanic forcing only. In this sense the impact of the different forcing will become more evident in the case that the authors want to extend the discussion on the role of the varying forcing.

iv) L167-172: I am concerned that the arbitrary choice of the base 5-year period prior to the corresponding eruptions shown in Fig. 2 might create a bias in the calculation of anomalies. It might work in many cases but it could be the reason why the Pinatubo eruption appears not to have an effect on the NED frequency. It could be that in the 5 years prior to the Pinatubo eruption, a strong ENSO event contaminates the 5-year base period. I suggest further analysis by using alternative 5-year periods or even a larger common period as a base to study all eruptions and specifically the Pinatubo eruption.

v) Overlaying the 20CR and L-days lines in Figs. 2a-c could be added for completeness.

vi) L162: What is an effusive eruption?

vii) L167: Could these runs be among the 5 CESM-LME with volcanic forcing only?

viii) L161-166: Why the high-latitude Laki eruption is expected to have a faster and short-lived influence? Is it related to the proximity of Iceland to the North Atlantic storm track that facilitates a faster spreading of the volcanic ash around the Northern Hemisphere? And why is it expected to be short-lived?

ix) L173: "see gray line in Figure 2" > "see gray line in Figure 1" ?

x) L174: Which signatures are inconsistent during the 21st century?

xi) L177: I cannot locate the Kuwai eruption in Fig. 3.

xii) L179: But in Fig. 2d above the reduction of NED frequency for +1 year is not present in CESM-LME (for which results are shown here in Fig. 3).

xiii) L179-182: This is a very interesting result. But what determines that the decline of Etesians will appear in the summer of the same or next year? Why the high-latitude eruption in the Northern Hemisphere are particularly effective in changing the summer circulation over the EMed?

5) **Waning of the Etesians after volcanic eruptions**

i) What are the units in the contours depicted in Fig. 4? Consistent with text the units should be hPa and not Pa. This should be clarified in the caption.

ii) L200-202: I absolutely agree with the inference that the low pressure system over the Middle East becomes shallower after major volcanic eruptions though I would not call it "Anatolian low" because it is not located over the Anatolian plateau. It is evident that the Etesians wane because the pressure gradient weakens over the EMed. But it is also evident that a swallower high pressure system over the Balkans has a contribution to the weakening pressure gradient over the EMed. Therefore the authors need to describe this contribution as well, otherwise the discussion of Fig. 4 is incomplete.

iii) L203-205: north-east > northeasterly wind anomalies? I do not agree that the anomalous flow has the same direction over the Arabian Sea and the Bay of Bengal. In anything, in Fig. 4a, arrows are directed in opposite direction over these two regions.

iv) L208-210: I do not see any anomalous northwesterly flow over the Arabian Sea in Fig. 4d. Interestingly, the anomalous flow is easterly in CESM-LME (Fig. 4d) but westerly in 20CR (Fig. 4e) over the Arabian Sea. Could the authors comment on this discrepancy?

6) **Waning of the monsoon-desert mechanism after volcanic eruptions**

i) L213-215 & Fig. 5: Contours and their units are not explained in the caption. Shades of blue in the contours appear to represent ascending motions and have negative values as expected (units Pa/s). But, unless I miss something, anomalies are expressed in units (-Pa/s), which is confusing.

ii) L215-217: The monsoon desert mechanism is not thought (by the studies mentioned here) to represent a closed circulation or overturning circulation (Walker-type circulation) with ascending motions over south Asia and descending motions over the EMed. If anything, Rodwell and Hoskins (1996) and Tyrlis et al (2013) present evidence corroborating the notion that the monsoon induces a zonal asymmetry that interacts with the mid-latitude westerlies resulting in enhanced subsidence over the EMed. See for example the discussion in pgs 1396-1397 in Rodwell and Hoskins (1996).

iii) L223-224: "as perhaps expected from the strongest decline in the Etesian winds". This inference is not clear to me. How is the decline in the Etesians related to the anomalies in ascent and descent over India and the EMed?

iv) L229-230: Actually there is no clear reduction of ascending motions over the "box region" in Fig. 5e. I can see a blue area over continental India & Bay of Bengal and a red area further to the south.

v) L237-241 & Fig. 6: Please consider carefully the description of units. In the caption units are referred to as (-Pa/s) while in the label of the horizontal axis as (Pa/s).

vi) L234-234: "Moreover, we identify an almost linear relationship between changes in ISM strength and NED anomalies". I am looking for a punch line in this paragraph describing the results shown in Fig. 6. Is it that there is a linear relationship between ISM strength and NED anomalies or that stronger volcanic eruptions can produce a stronger decline in the frequency Etesians? To put in another way, Fig. 6 is composed of years when volcanic eruptions occurred and it does not describe the climatological strength of the monsoon-desert mechanism, as inferred by the above sentence.

7) L245-247: Here the authors infer that cooling over the EMed is due to reduced adiabatic heating. Although that someone would expect that a reduction in the subsidence over the region would lead to a reduction of the adiabatic warming, evidence about this is not provided in any of the figures. Interestingly, in the Introduction (L33-34), it is mentioned that the Etesians have a cooling effect over the EMed and someone could expect that their decline would be associated with surface warming and not surface cooling, as depicted in Fig. 4a-c. More detailed analysis is required before inferences, such as the above, are reached.

8) L251-253: I do not agree that volcanic eruptions have a trivial effect on SLP over the Balkans. Figs 4a-c suggest that negative SLP anomalies prevail over Europe and expand towards the Balkans. This anomaly contributes to the weakening of the pressure gradient over the EMed. Thus, the decline of the Etesians is not caused only because the thermal low over the Middle East becomes weaker.

9) L18-20, L72 & L279-280: Could the authors comment on the dynamics that cause the weakening of the "Anatolian low"? As mentioned above, it is evident from Figs 4a-c that actually the impact of the volcanic eruption is the appearance of a dipole of SLP, with negative SLP anomaly over Europe and positive SLP over the Middle East.

**Minor comments**

L12-14: This is a long sentence and a bit difficult to understand, please rewrite.

L18: Late summer months?

L18-20: This is a long and complicated sentence. As mentioned above, I find it difficult to understand the dynamical link between the weakening of the monsoon-desert mechanism and the weakening of the Anatolian low.

L26: What does "Balkan high" refer to?

L27-29: Something is missing in this sentence. Please rephrase.

L30: What do the authors mean by "as they are synchronized with the summer monsoon"? With the Indian summer monsoon?

L30-31: What do the authors mean here by "synoptic system"?

L32-33: It is not clear how the Etesians are "amplified by a large-scale subsidence established in summer months under the influence of the Indian and Asian summer monsoon". Please clarify.

L37: "increased atmospheric blocking activity over Europe"

L42-43: What do the authors imply by "vice versa"? I think that this sentence could be removed.

L75-77: This is a confusing sentence. Please rewrite.

L91: Does "1" correspond to a footnote?

L119-122: Long and confusing sentence. Please rewrite.

L131: What is the meaning of "muted" and "punctuated" here?

L155: "In addition" - > "For completeness, "

L230: "in negligible" - > "is negligible"

L259: Do you mean here "anomalous temperature gradients"? Please clarify.

L261-262: I might be wrong. But isn't Pinatubo a volcano in the tropical region?

L271: "positive?"?

Caption of Fig. 1: "volcaninc"->"volcanic"

Caption of Fig. 2: Please add explanations for orange and red lines.

L12, 19, 23, 25, 44, 56, 73, 120 and possibly elsewhere: It may be correct to write "over the eastern Mediterranean" or "over the central Aegean".

L12 and elsewhere: It may be better to replace "Etesian winds" with "Etesians".

---

## Referee Comment (RC2)

**Review of Misios et al.** *"Decline of Etesian winds after large volcanic eruptions in the last millennium".*

This is an interesting and novel paper. I thank authors for doing this work and the editor for giving me opportunity to review it. Misios et al., using simulations from CESM-LME, show decline in strength and occurrence of summer Etesian winds following explosive volcanic eruptions. Such understanding will help improve weather predictions in the region. They show that warmer summer over India following eruptions weakened ISM that eventually declined Etesian winds. I have one major theoretical concern on their results on temperature anomalies post eruptions and how that would affect ISM. Please see my comments below.

**Major concerns**
**Figures 4 and 5**
I am surprised to see positive summer temperatures following a few eruptions over India. In Stevenson et al., 2016 (Figure 3), using CESM, they show negative tmp anomalies in winter for 1 and 2 years post eruptions when plotting for entire Tropical and Northern eruptions. In my experience (not published), also see negative anomalies in India for spring season as well for tropical and norther eruption composites. I have not looked at CESM tmp anomalies for summer season, so maybe it is a summer thing or maybe some eruptions have such effect.

If this is not too much work, I would like to see summer temperature anomaly plots for n=0, 1 and 2 years post eruption for each eruption and also a composite plot by eruption type (all Tropical and all Northern eruptions).

Are these anomalies in temperature, wind, and velocity for the year of eruption, or 1 or 2 years post eruptions? I think the authors say first summer post eruption, suggest making it clear in caption as well.

Furthermore, warmer temperatures in Indian land compared to the Indian Ocean region (as seen in Figure 4), would increase land-sea thermal gradient (that is increase ocean-to-land pressure gradient) and that is supposed enhance southwest ISM winds and rainfall. See, Ramesh and Goswami (2007) that show warmer ocean than land in past decades reduced the land-sea thermal gradient that declined the Indian monsoon rainfall. What is authors' take on this fact?

> Ramesh and Goswami, 2007. Reduction in temporal and spatial extent of the Indian summer monsoon. *Geophysical Research Letters.*

**Other minor comments**
**Introduction**
I very much liked the technical content of the Introduction. But I think it would be nice to have a short paragraph on the value of Etesian winds after its Introduction in first paragraph. So a second paragraph on: what changes in climate conditions we see due to changes in winds, and what regions and who (people) are affected by those changes, what are its socioeconomic effects with specific examples, and ending the paragraph with a note on why we need to study this wind system.

The authors give some hints on socioeconomic impacts, but do not provide specific examples at the end of first paragraph. Similarly, at the beginning of third paragraph, they give only mentioned the region is a climate change hotspot, but then quickly move forward on what they plan to do.

I would recommend making one separate paragraph on discussing the societal impacts of changes in Etesian winds. The authors can condense the last two paragraphs to make one to cover the increase in words in adding a new paragraph.

**Line 39-41** I would recommend against the use of word 'thought' when making statement based on published work that have proven the fact stated.

**Line 53-54** The mismatch exist maybe because the ISM has actually declined in the past few decades (Ramesh and Goswami, 2007; Kumar et al., 2020)

> Ramesh and Goswami, 2007. Reduction in temporal and spatial extent of the Indian summer monsoon. *Geophysical Research Letters.*
> Kumar et al., 2020. Recent unprecedented weakening of Indian summer monsoon in warming environment. *Theoretical and Applied Climatology.*

**Section 2.2 (labeled as 2.1)**
Can the authors add a sentence on how 'typical (common)' is that the wind is strongest at the selected grid of study (37.5N, 25.E)? Do the authors mean, observations show strongest winds at this particular location over certain period? I would recommend to be specific when the author say 'typical'.

And, maybe it is more appropriate to discuss model performance compared to observations/ reanalysis in the first section under Results.

**Line 131-132** Can you support your statement by doing the significance test for the trend over the past century? This result, positive trend in NED and WSP over the 20th century, does not align with monsoon behavior, which is declining over the past several decades (Kumar et al., 2020). I am sure this study is focused on volcanic forcing and not long-term trends, but still it makes sense to add a sentence on this discrepancy.

**Line 195** zonal mean temperatures subtracted from what? You mean subtracted from temperature 5 years prior to eruption?

**Conclusion** I like the way the authors have not only summarized their findings, but concluded with so what do we learn now which we did not know prior to this work.

---

## Author Comment (AC1)

**Review of Misios et al. "Decline of Etesian winds after large volcanic eruptions in the last millennium".**

This is an interesting and novel paper. I thank authors for doing this work and the editor for giving me opportunity to review it. Misios et al., using simulations from CESM-LME, show decline in strength and occurrence of summer Etesian winds following explosive volcanic eruptions. Such understanding will help improve weather predictions in the region. They show that warmer summer over India following eruptions weakened ISM that eventually declined Etesian winds. I have one major theoretical concern on their results on temperature anomalies post eruptions and how that would affect ISM. Please see my comments below.

General Comments

We would like to thank both reviewers for their thoughtful, encouraging, and constructive comments. Reviewer 1 provided a very detailed list of comments, emphasizing the mechanism and the relation to the ISM. Reviewer 2 acknowledges the implications of our work in improving seasonal prediction of the summer circulation in the Mediterranean region and expressed concerns on the modelled effects in the ISM region. Our reply to Reviewer 2 demonstrates that a) the summer warming over India is a robust feature in the CESM-LME simulations for the strongest eruptions at least. This evidence is further supported by independent model studies. 20CR also demonstrates a summer warming over India after Pinatubo. We also demonstrate that this is a summer season affect, as speculated; CESM-LME shows a cooling in the post-eruption winter, consistent with previous studies.

Key changes in the revised manuscript

- We include a new Figure 1. The figure numbering in the original version thus has increased by one. The new figure 1 is supplementary Figure 1, with the addition of displaying the climatology of SLP and surface winds in JA for the CESM-LME, as recommended by Reviewer 2.
- As recommended by Reviewer 1, in Figure 2 (old Figure 1) we have merged all-forcing and volcanic forcing times series. Hopefully, this simplifies the reading and improves the clarity of the manuscript.
- We have updated the discussion of Figure 5 (old figure 4). We have added a new supplementary figure 2 where we present surface temperature anomalies after subtracting the zonal mean value. This emphasizes the dynamic response of surface temperature anomalies. In the EMed, the amplified cooling is related to the reduced descending motion (as manifested by Omega velocity anomalies in Figure 6), and hence less adiabatic heating.
- Reversed color scaling is used in Figure 6 to be consistent with the units of Omega velocity (-Pa/s).
- New Figure 7 is updated with new Omega velocity units (-Pa/s).

Below, we reply to each comment and describe the changes to the manuscript resulting from them. (*new or revised text in italics*)

**Major**                                            **concerns**
**Figures**                   **4**                           **and**                           **5**

I am surprised to see positive summer temperatures following a few eruptions over India. In Stevenson et al., 2016 (Figure 3), using CESM, they show negative tmp anomalies in winter for 1 and 2 years post eruptions when plotting for entire Tropical and Northern eruptions. In my experience (not published), also see negative anomalies in India for spring season as well for tropical and norther eruption composites. I have not looked at CESM tmp anomalies for summer season, so maybe it is a summer thing or maybe some eruptions have such effect.

If this is not too much work, I would like to see summer temperature anomaly plots for n=0, 1 and 2 years post eruption for each eruption and also a composite plot by eruption type (all Tropical and all Northern eruptions).

**Reply**: Yes, indeed Stevenson et al paper shows negative temperature anomalies over India but for the winter months, as the reviewer correctly mentions. Our analysis focuses on the summer months, and this explains the difference. To start with, the independent work of Dogar and Sato (2019) shows positive temperature anomalies in the summer season both in an observational product and high-resolution simulations with a different model (see their figure 1). The contrast between the DJF and JJA temperature anomalies over India is also implied (India is marginally covered) in figure 3 in Dogar, Stenchikov et al. (2017), with warming anomalies in the summer season (cooling in winter) in year +2 after the El Chichon and Pinatubo eruptions. This suggest that temperature anomalies over India are strongly dependent on the season, and a possible explanation is found in the summer cloud cover anomalies. A reduced cloud cover in summer related to a weakened ISM and an increased direct SW heating should warm the surface.

Prompted by the comment, we have calculated winter (DJF) temperature anomalies in the first post eruption year after Samalas, Laki Tambora and Pinatubo (January gives the year). This figure should be compared with (new) Figure 5. We find cooling anomalies over India as suggested by the reviewer. This confirms our argument that a warmer India is a summer-time feature.

[Figure]

*Reply Fig. 1 Anomalies of surface temperature (K, shaded), 10m winds (m/s, arrows) and SLP (Pa, contours) in the post-eruption winters (DJF) following Samalas, Laki, Tambora and Pinatubo.*

Are these anomalies in temperature, wind, and velocity for the year of eruption, or 1 or 2 years post eruptions? I think the authors say first summer post eruption, suggest making it clear in caption as well.

**Reply**: These anomalies refer to the year of the maximum decline of etesian winds. For the tropical volcanoes, this is +1 year while for the high-latitude volcanoes this is at the eruption year. We suspect that the rather similar time lags for every tropical and high-latitude eruption might be related to the way the volcanic forcing is implemented in the model. This is commented in Section 4 *"Eruptions of unknown dates are assumed to begin in April and peak in June-July, a simplification that leads to very similar time evolution of the volcanic forcing (Stevenson, Brady et al. 2016). This might lead to an overestimation of the volcanic forcing in the summer months for some of the unknown eruptions."*

Furthermore, warmer temperatures in Indian land compared to the Indian Ocean region (as seen in Figure 4), would increase land-sea thermal gradient (that is increase ocean-to-land pressure gradient) and that is supposed enhance southwest ISM winds and rainfall. See, Ramesh and Goswami (2007) that show warmer ocean than land in past decades reduced the land-sea thermal gradient that declined the Indian monsoon rainfall. What is authors' take on this fact?

Ramesh and Goswami, 2007. Reduction in temporal and spatial extent of the Indian summer monsoon. *Geophysical Research Letters.*

Reply: We think the key mechanisms are those described in Stevenson et al. The model simulates a cyclonic wind pattern over the Arabian sea which weakens the Somali jet. Moreover, the model shows a tendency for positive temperature anomalies in the Equatorial Pacific suggesting a response similar to Positive ENSO. This should impact the Walker circulation by reducing the upward motions over Indian and Southern Asia. This is the mechanism proposed by Kumar, Rajagopalan et al. (1999) Reduced cloud cover and increased SW heating should result in the surface warming evident in post-eruption summers as simulated with the CESM-LME.

**Other                                                       minor                                                    comments**
**Introduction**
I very much liked the technical content of the Introduction. But I think it would be nice to have a short paragraph on the value of Etesian winds after its Introduction in first paragraph. So a second paragraph on: what changes in climate conditions we see due to changes in winds, and what regions and who (people) are affected by those changes, what are its socioeconomic effects with specific examples, and ending the paragraph with a note on why we need to study this wind system.

The authors give some hints on socioeconomic impacts, but do not provide specific examples at the end of first paragraph. Similarly, at the beginning of third paragraph, they give only mentioned the region is a climate change hotspot, but then quickly move forward on what they plan to do.

I would recommend making one separate paragraph on discussing the societal impacts of changes in Etesian winds. The authors can condense the last two paragraphs to make one to cover the increase in words in adding a new paragraph.

**Reply:** This is a very useful comment. Indeed, Etesian winds are, as we mention in the last paragraph of the Summary and conclusions, *"important for ecosystem services, wildfire prevention, air quality forecasts, tourism, energy production and economic development (Athanasopoulou, Protonotariou et al. 2015, Dafka, Toreti et al. 2018)"*. The suggestion of the Reviewer, however, requires a dedicated paragraph which we fear will defer readers from the main focus of the paper. We have added one more reference at the very end of the first paragraph

of introduction and we encourage the interested reader to find many additional studies in those papers "*thus regulating summertime conditions and affecting several environmental and socio-economic sectors (e.g. Athanasopoulou, Protonotariou et al. 2015, Dafka, Toreti et al. 2018 and references therein).* "

**Line 39-41** I would recommend against the use of word 'thought' when making statement based on published work that have proven the fact stated.

**Reply**: Now it reads "*… and is amplified by …*"

**Line 53-54** The mismatch exist maybe because the ISM has actually declined in the past few decades (Ramesh and Goswami, 2007; Kumar et al., 2020)

Ramesh and Goswami, 2007. Reduction in temporal and spatial extent of the Indian summer monsoon. *Geophysical Research Letters.*

Kumar et al., 2020. Recent unprecedented weakening of Indian summer monsoon in warming environment. *Theoretical and Applied Climatology.*

**Reply**: Thank you for this comment. We have changed the sentence to *"Such positive trends in model simulations, however, are not supported by observations of Etesians over the past decades (Poupkou, Zanis et al. 2011, Rizou, Flocas et al. 2018) and negative trends in Etesians could be associated with a weakening of the ISM in the past few decades (Kumar, Naidu et al. 2020)."*

**Section 2.2 (labeled as 2.1)**

Can the authors add a sentence on how 'typical (common)' is that the wind is strongest at the selected grid of study (37.5N, 25.E)? Do the authors mean, observations show strongest winds at this particular location over certain period? I would recommend to be specific when the author say 'typical'.

**Reply**: We mean that the strongest wind in this location is found in July and August. We have rephrased the sentence "*so as to select the months typically demonstrating the strongest wind speeds under the influence of monsoon convection over northern India (Tyrlis and Lelieveld 2013)*"

And, maybe it is more appropriate to discuss model performance compared to observations/ reanalysis in the first section under Results.

**Reply**: We have included a new Figure 1 (essentially the old Supplementary Figure 1) and some additional discussion in Section 2.2 to address this suggestion. The model performance in relation to ENSO variability is discussed in Section 4, "*The CESM model also suffers from over-active ENSO variations compared to the observations, which need to be considered when disentangling direct volcanic effects and ENSO.*"

**Line 131-132** Can you support your statement by doing the significance test for the trend over the past century? This result, positive trend in NED and WSP over the 20th century, does not align with monsoon behavior, which is declining over the past several decades (Kumar et al., 2020). I am sure

this study is focused on volcanic forcing and not long-term trends, but still it makes sense to add a sentence on this discrepancy.

**Reply:** A very valid point. We have tested the significance of the trends using the non-parametric Mann-Kendall test (implemented in ncl language as trend_manken function). Over the 1800-2005 period, we find a Theil-Sen estimate of linear trend in the NED for all-forcing CESM-LME runs of 0.018 days/year (p=1), while over the 1900-2005 period it is 0.025 days/year (p=0.99). In contrast, the volcanic-only ensemble shows no trends for both periods ( 0 days/year). This means that the CESM-LME supports our statement about *"an increasing number of Etesian days and WSP in the all-forcing simulation"*.

However, our choice of presenting all-forcing and volcanic-only forcing time series in Figure 1 separately might add some level of confusion to the readers, as pointed out by Reviewer 1. So, in the revised text we decided to merge all-forcing and volcanic-only forcing (black and cyan lines in old figure 1) in a single line representing a grant ensemble of 17 realizations. This is valid given the focus of the paper on volcanoes and not on trends. For this reason, the sentences *"Periods of muted interannual variability are punctuated by periods of enhanced activity, with some evidence for positive trends over the last century, suggestive of an increasing number of Etesian days and WSP in the all-forcing simulation. This is consistent with independent model simulations of future greenhouse gas emissions resulting in higher WSPs and NEDs."* are modified and moved to Section 4: Discussion and Summary.

**Line 195** zonal mean temperatures subtracted from what? You mean subtracted from temperature 5 years prior to eruption?

**Reply:** No, at the chosen year after the eruption we remove the zonal mean value. This comment motivated us to change the text accordingly to facilitate an easier reading. In the revised version, we present temperature anomalies without subtracting zonal mean. Additionally, we have added a new supplementary figure 2, where we present surface temperature anomalies after subtracting the zonal mean value.

**Conclusion** I like the way the authors have not only summarized their findings, but concluded with so what do we learn now which we did not know prior to this work.

**Reply:** Thank you.

References

Athanasopoulou, E., A. P. Protonotariou, E. Bossioli, A. Dandou, M. Tombrou, J. D. Allan, H. Coe, N. Mihalopoulos, J. Kalogiros, A. Bacak, J. Sciare and G. Biskos (2015). "Aerosol chemistry above an extended archipelago of the eastern Mediterranean basin during strong northern winds." Atmospheric Chemistry and Physics **15**(14): 8401-8421.
Dafka, S., A. Toreti, J. Luterbacher, P. Zanis, E. Tyrlis and E. Xoplaki (2018). "Simulating Extreme Etesians over the Aegean and Implications for Wind Energy Production in Southeastern Europe." Journal of Applied Meteorology and Climatology **57**(5): 1123-1134.
Dogar, M. M. and T. Sato (2019). "A Regional Climate Response of Middle Eastern, African, and South Asian Monsoon Regions to Explosive Volcanism and ENSO Forcing." Journal of Geophysical Research-Atmospheres **124**(14): 7580-7598.

Dogar, M. M., G. Stenchikov, S. Osipov, B. Wyman and M. Zhao (2017). "Sensitivity of the regional climate in the Middle East and North Africa to volcanic perturbations." Journal of Geophysical Research-Atmospheres **122**(15): 7922-7948.

Kumar, K. K., B. Rajagopalan and M. A. Cane (1999). "On the weakening relationship between the indian monsoon and ENSO." Science **284**(5423): 2156-2159.

Kumar, P. V., C. V. Naidu and K. Prasanna (2020). "Recent unprecedented weakening of Indian summer monsoon in warming environment." Theoretical and Applied Climatology **140**(1-2): 467-486.

Poupkou, A., P. Zanis, P. Nastos, D. Papanastasiou, D. Melas, K. Tourpali and C. Zerefos (2011). "Present climate trend analysis of the Etesian winds in the Aegean Sea." Theoretical and Applied Climatology **106**(3-4): 459-472.

Rizou, D., H. A. Flocas, M. Hatzaki and A. Bartzokas (2018). "A Statistical Investigation of the Impact of the Indian Monsoon on the Eastern Mediterranean Circulation." Atmosphere **9**(3).

Stevenson, S., E. Brady, J. Fasullo, B. Otto-Bliesner and S. Stevenson (2016). ""El Niño Like" Hydroclimate Responses to Last Millennium Volcanic Eruptions." Journal of Climate **29**(8): 2907-2921.

Tyrlis, E. and J. Lelieveld (2013). "Climatology and Dynamics of the Summer Etesian Winds over the Eastern Mediterranean." Journal of the Atmospheric Sciences **70**(11): 3374-3396.

---

## Author Comment (AC2)

Review comments on the submitted article "**Decline of Etesian winds after large volcanic eruptions in the last millennium**" by Misios S, Logothetis I, Knudsen M F, Karoff C, Amiridis V and Tourpali K.

**General comments**

This study uses data from the 20$^{th}$ Century Reconstruction project (20CR) and the output from the Last Millennium Ensemble simulations with the model CESM (CESM-LME) to investigate the impact of large volcanic eruptions on the summer circulation over the eastern Mediterranean (EMed) and, in particular, the persistent northerly flow, known as the Etesians. The authors present evidence that major volcanic eruptions in the Northern Hemisphere are able to weaken the pressure gradient over the EMed, reduce the intensity of the Etesians and their frequency of appearance. It is also shown that the well-known monsoon-desert mechanism weakens during summers after major eruptions. The effect of eruptions appears to be more obvious for eruptions before the 20$^{th}$ century when the volcanic radiative forcing is considered to be stronger. The work bears the necessary scientific merit and I reckon that it can have an important contribution to the topic because it is characterized by the element of novelty. However, the manuscript suffers at the writing and presentation level. Frequently, there is incomplete discussion on the results, the uncertainties involved and the dynamics explaining the aforementioned link. My recommendation is that the manuscript could only be ready for publication by Weather and Climate Dynamics after major revisions, mainly due to the large number of suggested comments. Below I make specific suggestions that aim at improving the manuscript.

General Comments

We would like to thank both reviewers for their thoughtful, encouraging, and constructive comments. Reviewer 1 provided a very detailed list of comments, emphasizing on the mechanisms and particularly the relation to the ISM. Reviewer 2 acknowledges the implications of our work in improving seasonal prediction of the summer circulation in the Mediterranean region but expressed concerns on the modelled effects in the ISM region. Our reply to Reviewer 1, addresses in detail comments about the presentation quality, simplifies the methodology and the definition of CESM-LME, extends the description of the datasets and simulations, addresses some ambiguities in the description of our results and elaborates on the mechanisms in relationship to the ISM. We have to acknowledge the reviewer #1 for providing such a detailed and spot-on comments and we hope that the reviewer will find our revised manuscript considerably improved.

Key changes in the revised text

- We include a new Figure 1. The figure numbering in the original version thus has increased by one. The new figure 1 is supplementary Figure 1, with the addition of displaying the climatology of SLP and surface winds in JA for the CESM-LME, as recommended by Reviewer 2.
- As recommended by Reviewer 1, in Figure 2 (old Figure 1) we have merged all-forcing and volcanic forcing times series. Hopefully, this simplifies the reading and improves the clarity of the manuscript.

- We have updated the discussion of Figure 5 (old figure 4). We have added a new supplementary figure 2 where we present surface temperature anomalies after subtracting the zonal mean value. This emphasizes the dynamic response of surface temperature anomalies. In the EMed, the amplified cooling is related to the reduced descending motion (as manifested by Omega velocity anomalies in Figure 6), and hence less adiabatic heating.
- Reversed color scaling is used in Figure 6 to be consistent with the units of Omega velocity (-Pa/s).
- New Figure 7 is updated with new Omega velocity units (-Pa/s).

Below, we reply to each comment and describe the changes to the manuscript resulting from them. (*new or revised text in italics*)

**Specific                                                                                          Comments**
**1) Presentation of the coincidence between volcanic eruptions and reductions in the frequency of Etesians**:

The presentation of results in Figure 1 should to be improved:

  i)      Overlaying the cyan and black lines (Figs. 1a and b) that represent CESM-LME all forcing ensemble and CESM- LME volcanic-only forcing makes it difficult to distinguish between the two. Perhaps I miss something but actually, in the text there is very limited discussion on the difference between the two that draws conclusions on the importance of the volcanic forcing alone compared to the all-forcing simulations. If so, does it make sense to show here only one curve? Otherwise, please comment on the difference between the two and the necessity of showing both of them here.

**Reply:** In the revised text we added L.139 *"forcings (greenhouse gases, solar variability, volcanic, land use and orbital)"*. The reason that we kept the two ensembles separate in the original text was mainly prompted by the evidence that they depart from each other after 1900, indicative of an influence from the GHG forcing. This has been also commented by Reviewer 2 asking about the significance in the trends found after 1900 in the all-forcing ensemble. We have tested the significance of the trends using the non-parametric Mann-Kendall test (implemented in ncl language as trend_manken function). Over the 1800-2005 we find a Theil-Sen estimate of linear trend in the NED for all-forcing CESM-LME runs of 0.018 days/year (p=1) while over the 1900-2005 period is 0.025 days/year (p=0.99). In contrast, the volcanic-only ensemble shows no trends for both periods ( 0 days/year). This means that the CESM-LME supports our statement about *"an increasing number of Etesian days and WSP in the all-forcing simulation"*.

However, our choice of presenting all-forcing and volcanic-only forcing time series in (old) Figure 1 separately may confuse readers, as pointed by Reviewer 1. In the revised text we decided to merge all-forcing and volcanic-only forcing (black and cyan lines in old Figure 1) in a single line representing a grant ensemble of 17 realizations. This is valid given the interest of the paper in volcanoes and not in trends.

ii)      Caption and description of Fig. 1c are problematic. It is not clear what the red and black lines represent. I can only speculate that the black line is the same as the black line (?) in panels 1a,b. But again the vertical range is suppressed and the line appears to very flat, which hinders the effort to draw conclusions on the timing with the volcanic eruptions.

**Reply:** Old Figure 1c shows NED time series in CESM-LME, 20CR and L-days using the same range in the Y axis. Yes, we should expect a reduction of the variance of NED by the ensemble averaging. For this reason, the variance in CESM-LME should not be compared to the variance of the observations, which under an "ensemble" thinking, it can be considered as a single realization. We agree with the reviewer that it is difficult to draw conclusions for any eruption after 1836, except perhaps the Krakatoa 1883. We note that the black line should not be used to infer the timing of eruptions because the volcanic signal is super imposed to natural variability. Grey lines in (new) Figure 2 and Table 1 are giving the time of eruptions.

iii)     Grey line gives the outgoing clear sky SW radiation in the Northern Hemisphere, but units are not given here except that "scaling is relative". Perhaps units should be also given on the right hand side vertical axis.

**Reply**: Units should be Watts/m$^2$. To avoid adding too many numbers we have chosen a relative scale. We consider the absolute value irrelevant here and does not add to the discussion. It is only used for visual purposes in (new) Figure2 to highlight that peaks in the forcing coincide with drops in the Etesian winds.

iv)      L135, L151 and caption of Fig. 1: More on the previous comment. NH, SW and TOA are not explained here. Could you please give more information about the physical basis that relates volcanic eruptions and coincident higher outgoing SW? Does "outgoing clear sky SW radiation" refer to radiation reflected by aerosols in cloud-free regions?

**Reply**: Yes this is it. Outgoing also include the surface reflection but given the short time aspect of our analysis, it should be considered unchanged. So essentially, we capture the influence of the aerosols in the stratosphere. An alternative choice could be using globally averaged SW radiation but we find more appropriate to present NH averaged time series as we focus in the NH.

v)       I find extremely hard to locate the dates of volcanic eruptions, which correspond to peaks in the grey line (Fig. 1) so as to check the coincidence with the weakening of the Etesians over the EMed. The authors should add several vertical lines that mark all major eruptions mentioned in Table 1. A solution could be to add an initial letter (i.e., P: Pinatubo) that should be also indicated in Table 1.

**Reply**: We think that dates of eruptions can be taken from Table 1.

2) **Details about CESM-LME simulations.**

i) L82-85: Here the authors describe briefly the sets of simulations used but they do not disclose details on how the initialization of the various ensemble members was performed.

What is the meaning of "twin ensemble of 5 members"? How was the initialization of these ensemble members made?

**Reply**: Thanks for this comment. We have added some extra description for CESM-LME. As described in the Otto-Bliesner et al paper (the reference is given in our text), the initialization is made by adding a small random roundoff (order $10^{-14}$ °C) differences in the air temperature field at the start of each ensemble member. Perhaps, this is too technical information and it can be avoided in our description of the CESM. The interested reader should be looking for these details in the reference paper.

ii) Also, it may be helpful that the authors use different acronyms for results obtained with the use of simulations from CESM-LME all forcing, only volcanic forcing simulations and for the cases that all 17 members of CESM-LME are grouped together. For example something like, CESM-LMEaf, SESM-LMEvf and CESM-LMEt, respectively. These acronyms could be used consistently throughout the manuscript to alleviate confusion.

**Reply**. This comment is similar to a previous comment and points to the overall confusing description of the (old) Figure 1. We changed the text and now we use CESM-LME to describe the 17-member ensemble with the CESM model.

iii) Details of the spatial resolution of the CESM-LME simulations and the 20CR dataset should be also listed. In L125-126, the authors argue that PDF of NED is skewed towards higher values in 20CR when compared to the one corresponding to CESM-LME. But it is not explained how the resolution is expected to affect the PDF. Please clarify.

**Reply:** We have added details about the resolution in the model and 20CR. Increased resolution should better simulate topography and land-sea contrast which should increase the representation of "channeling" of the Etesian winds in the Aegean Sea. This should influence the wind speed climatology in the summer months as well as wind bursts, which should strengthen. For example, in an unpublished work with two different resolutions of the ECHAM5 model, we find much improved winds in EMed when going from 310 Km to 210 Km grid size.

Wind speed climatology in the CESM model with 1° grid size reasonably compares with the 20CR, as shown in the (new) Figure 1b, but underestimates on higher wind speeds. This can also be seen in the following figure where we have included the PDF of ERA5 reanalysis over the admittedly shorter period (1979-2019, green line, CESM-LME black, 20CR red). As expected, the mean value again is higher in ERA5 than CESM-LME, again because of the finer grid size.

[Figure]

Reply Fig. 1 Probability density functions of northerly JA WSP for CESM-LME (black), 20CR (red) and ERA5(green). Dash lines denote the median values.

iv) L157-158 & 270-272: I am not sure that the ensemble size of 17 members is adequate to ensure that volcanic signature will emerge from natural variability. And even more so with respect to the effect of the ENSO on ISM. In recent studies, there is a tendency for the use of sets of at least 100 ensemble members to disentangle signal form natural variability.

**Reply**: We are on the same page here. All ensemble runs agree on the response in the case of the strongest Samalas eruption but there is a considerable spread for less severe eruptions. Pinatubo is a good example here because we don't find a significant and consistent signal. We have tested the Pinatubo signature in another large ensemble of the same CESM (CESM large ensemble Community Project, https://www.cesm.ucar.edu/projects/community-projects/LENS/) which considers 42 runs from 1920 to 2005 (See figure below). We find a weak reduction of NED in the ensemble mean anomaly (thick orange line, individual runs with thin orange lines), which might be significant. We think that even larger ensembles are needed to get a significant response for the Pinatubo eruption. It was not possible to test this with the MPI-ESM 100-member ensemble (MPI-GE) because no daily wind data have been archived.

[Figure]

*Reply Fig. 2 Percentage anomalies of NED for Pinatubo eruptions from ensemble mean CESM-GE (orange), 20CR (blue), L-days (green) and ERA5 (black), from 5 pre-eruption to 5 post-eruption years.*

vi)     L89-90 & L92: Is the volcanic forcing applied to the simulations available so as to confirm the hypothesis that the historical forcing was stronger? The terms "historical" and "recent" are frequently used in the manuscript, but sometimes very loosely. Please clarify.

**Reply**: The word "Historical" is a bit vague and it is changed to "pre-1900", to be precise. Grey lines in the new Figure 2 provide the visual justification that the 'pre-1900' eruptions are much stronger than the recent ones and a detailed discussion is given in  Sigl, Winstrup et al. (2015)

Bollasina, M. and S. Nigam (2011). "The summertime "heat" low over Pakistan/northwestern India: evolution and origin." Climate Dynamics **37**(5): 957-970.
Sigl, M., M. Winstrup, J. R. McConnell, K. C. Welten, G. Plunkett, F. Ludlow, U. Büntgen, M. Caffee, N. Chellman, D. Dahl-Jensen, H. Fischer, S. Kipfstuhl, C. Kostick, O. J. Maselli, F. Mekhaldi, R. Mulvaney, R. Muscheler, D. R. Pasteris, J. R. Pilcher, M. Salzer, S. Schüpbach, J. P. Steffensen, B. M. Vinther and T. E. Woodruff (2015). "Timing and climate forcing of volcanic eruptions for the past 2,500 years." Nature **523**(7562): 543-549.

Toohey, M., K. Krüger, H. Schmidt, C. Timmreck, M. Sigl, M. Stoffel and R. Wilson (2019). "Disproportionately strong climate forcing from extratropical explosive volcanic eruptions." Nature Geoscience **12**(2): 100-107.

Tyrlis, E. and J. Lelieveld (2013). "Climatology and Dynamics of the Summer Etesian Winds over the Eastern Mediterranean." Journal of the Atmospheric Sciences **70**(11): 3374-3396.

3) **Detailed discussion on the synchronicity between volcanic eruptions and decline in the frequency of Etesians in Fig. 1c**:

i) L142: As mentioned above, the meaning of the dark and red lines is not clarified. I speculate that the red line represents NED frequency based on 20CR.

**Reply**: This is related to a previous comment. We hope it reads much easier in the revised manuscript.

ii) L144: I disagree with the statement that both 20CR and CESM-LME do not show any significant decline after Krakatau (1883). Perhaps I miss something but I see many minima both in black and red lines (especially the red line – 20CR) in the 1890s, 1910s, 1930s, 1970s & 2000s. This is actually confirmed in the next sentence that refers to the eruption of Pinatubo, which is confusing.

**Reply**: For 20CR and L-days, the minimum is found a year earlier (1882) before the Krakatau eruption. CESM does show a reduction of about 20% (see new Sup. Figure 1) but it's hardly significant. For the multiple minima in the observations, we agree with the reviewer that other factors (e.g. ENSO) plus noise contribute the observed interannual variability of NED.

iii) L149: Is it possible to check whether the forcing is actually realistic in CESM-LME simulations specifically during this eruption?

**Reply**: This a pertinent question but difficult to be answered because of many unknowns regarding the eruption characteristics (e.g. emitted mass, emission maximum height). As described in the text, we have compared the NH upward clear-sky SW radiation at TOA between the CESM-LME and the 20CR. And we find "*about x5 stronger anomalies in the 20CR compared with CESM-LME, further supporting this possibility*". However, it might also be possible that 20CR overestimates the forcing. As far as we know, Oman et al., 2005 have simulated the effects of Katmai and they reported stronger summertime cooling than in CESM-LME, which also corroborates for a weak volcanic forcing in the CESM model.

iv) L153: This is very true for 20CR and L-days but may be not so accurate for CESM-LME. But again the vertical axis range used to plot the black line in Fig. 1c should be different so as to depict finer variability. Now the line appears rather flat.

**Reply**: This comment refers to (new) Figures 1 a and b and is similar to a previous comment. Over the last 150 years, CESM-LME shows little evidence for reduced Etesians in the aftermath of volcanic eruptions.

v) Surprisingly, there are many other minima in the red line that are not associated with volcanic eruptions. See for example the one a few summers prior to the 1883 Krakatau eruption. While many maxima in the grey line are not associated with minima in NED frequency, such as the case of early 1960s (1964?). This points out the relevance of other dynamical mechanisms (both in the tropics and mid-latitudes) that influence the summer circulation over the EMed.

**Reply**: That's very true. Etesians varying in interannual time scales and an overview of our current understanding is given in the Introduction. Our work provides evidence, for the first time, that the volcanic forcing is yet another factor influencing Etesian winds. This might be important for seasonal prediction.

4) **Discussion of the "epoch analysis" in Fig. 2**:

i) L154-155: Have you calculated the anomalies shown with respect to each ensemble separately?

**Reply**: Yes, we first calculate anomalies and the we then apply ensemble averaging. However, this is a linear analysis and results are independent of the order of operators.

ii) L156-157: In the text the authors mention that 17 thin grey lines represent the evolution for all 17 ensemble members. But in the caption of Fig. 2, it reads that results are shown for the 12 ensemble members of CESM- LME all forcing. Please clarify.

**Reply**: Thank you for this comment. The true number is 17 and this has been changed in the revised text.

iii) L157: It could be helpful to draw with different colors the curves representing the anomalies for the 5 CESM- LME members that use volcanic forcing only. In this sense the impact of the different forcing will become more evident in the case that the authors want to extend the discussion on the role of the varying forcing.

**Reply**. This could be an option, but we think that is valid to deal all 17 ensembles equally regarding the volcanic forcing. There is not any justifiable reason to assume different responses preconditioned by the additional forcing considered in the all-forcing ensemble, particularly prior the 1900 when the GHGs did not changed much.

iv) L167-172: I am concerned that the arbitrary choice of the base 5-year period prior to the corresponding eruptions shown in Fig. 2 might create a bias in the calculation of anomalies. It might work in many cases but it could be the reason why the Pinatubo eruption appears not to have an effect on the NED frequency. It could be that in the 5 years prior to the Pinatubo eruption, a strong ENSO event contaminates the 5-year base period. I suggest further analysis by using alternative 5-year periods or even a larger common period as a base to study all eruptions and specifically the Pinatubo eruption.

**Reply**: This is a common practice used in many earlier studies but we agree with the reviewer about the length of pre-eruption averaging. This is certainly a shortcoming in observations, as

pre-eruption 5 year averages could be biased by e.g a strong ENSO. However, we don't think that changes much our results from CESM-LME because of the ensemble averaging.

We nevertheless we have examined the recommendation and increased the pre-eruption period to 10 years. The following figure is similar to (new) Sup. Figure 1 but calculating anomalies considering 10 years before each eruption. As expected, our results from CESM-LME do not change.

[Figure]

Reply Fig. 3 Similar to (new) Sup. Figure 1 but considering 10 years before each eruption to calculate anomalies.

v) Overlaying the 20CR and L-days lines in Figs. 2a-c could be added for completeness.

**Reply:** It's not possible for those early eruptions because no daily wind observations are available. Reconstructions have monthly time resolution.

vi) L162: What is an effusive eruption?

**Reply**: From Wikipedia: An effusive eruption is a type of volcanic eruption in which lava steadily flows out of a volcano onto the ground. This contrast to a Plinian eruption (e.g. Samalas, Tambora, etc) which is explosive.

vii) L167: Could these runs be among the 5 CESM-LME with volcanic forcing only?

Reply: There is no physical reason for this.

  vii)     L161-166: Why the high-latitude Laki eruption is expected to have a faster and
           short-lived influence? Is it related to the proximity of Iceland to the North Atlantic
           storm track that facilitates a faster spreading of the volcanic ash around the
           Northern Hemisphere? And why is it expected to be short-lived?

Reply: it is short-lived because it does not travel in the stratosphere from tropics to the pole
with the Brewer-Dobson circulation as in the case of Tropical eruptions.

  viii)    L173: "see gray line in Figure 2" > "see gray line in Figure 1" ?

  Reply:Changed

x) L174: Which signatures are inconsistent during the 21$^{st}$ century?

Reply: It refers to NED anomalies. Changed to "while signatures are insignificant over"

xi) L177: I cannot locate the Kuwai eruption in Fig. 3.

Reply: It is the one in (new) Figure 4 a) showing a negative peak at Lag+1.

xii) L179: But in Fig. 2d above the reduction of NED frequency for +1 year is not present in
CESM-LME (for which results are shown here in Fig. 3).

Reply: We are not sure about the meaning of this comment. If the reviewer implies the
response of Kuwae, this is shown in the (new) Figure 4a.

xiii) L179-182: This is a very interesting result. But what determines that the decline of
Etesians will appear in the summer of the same or next year? Why the high-latitude eruption
in the Northern Hemisphere are particularly effective in changing the summer circulation over
the EMed?

Reply: Thank you for this comment. We think the time delay should be attributed to the
stratospheric circulation and particularly the time it takes volcanic aerosols to be transported
to NH stratosphere. Of course, this is much faster processes than a year but as NH
stratospheric warming builds with some months delay the strongest effects are simulated in
the following year. It also matters that most unknown eruptions in the last millennium in the
CESM model are assumed in spring. It terms of the increased sensitivity to NH eruptions, in
some sense, the dilution matters. Stratospheric aerosols from NH eruptions stay in the NH
whereas tropical eruptions transport aerosols in both hemispheres. For this reason,
stratospheric heating and anomalous zonal temperature gradients are often stronger for high
latitude eruptions. For example Toohey, Krüger et al. (2019) find "the Northern Hemisphere
extratropics up to 80% greater than tropical eruptions, as decreases in aerosol lifetime are

overwhelmed by the enhanced radiative impact associated with the relative confinement of aerosol to a single hemisphere."

5)  **Waning of the Etesians after volcanic eruptions**
i) What are the units in the contours depicted in Fig. 4? Consistent with text the units should be hPa and not Pa. This should be clarified in the caption.

**Reply**: We have added the units in the caption of new Fig 5.

ii) L200-202: I absolutely agree with the inference that the low pressure system over the Middle East becomes shallower after major volcanic eruptions though I would not call it "Anatolian low" because it is not located over the Anatolian plateau. It is evident that the Etesians wane because the pressure gradient weakens over the EMed. But it is also evident that a swallower high pressure system over the Balkans has a contribution to the weakening pressure gradient over the EMed. Therefore the authors need to describe this contribution as well, otherwise the discussion of Fig. 4 is incomplete.

Reply: We have changed the manuscript. We agree, Anatolian low increases but the strongest increase in found further south. Regarding the response over the high-pressure region, we find that it is much weaker. The following figure shows SLP anomalies over an extended domain. In the case of Samalas and Laki negative SLP anomalies are found more in the Central/Western Europe. In all cases the amplitude of positive SLP anomalies in the EMed is higher than the negative ones in the Balkans. For this reason, we argue in the text that the main influence comes through the low-pressure system.

[Figure]

*Reply Fig. 4 Replication of the (new) figure 5 but only for SLP anomalies (Pa) and over an extended domain.*

iii) L203-205: north-east > northeasterly wind anomalies? I do not agree that the anomalous flow has the same direction over the Arabian Sea and the Bay of Bengal. In anything, in Fig. 4a, arrows are directed in opposite direction over these two regions.

**Reply**: Changed. We agree with reviewer, and we kept just "Arabian Sea".

iv) L208-210: I do not see any anomalous northwesterly flow over the Arabian Sea in Fig. 4d. Interestingly, the anomalous flow is easterly in CESM-LME (Fig. 4d) but westerly in 20CR (Fig. 4e) over the Arabian Sea. Could the authors comment on this discrepancy?

**Reply**: It's difficult to interpret the response in 20CR and perhaps we shouldn't, because it's low significance. It looks as if there is a cyclonic circulation but a) it is much smaller and b) centered northward compared to CESM-LME. However, we think that this comment deserves a separate study.

6) **Waning of the monsoon-desert mechanism after volcanic eruptions**

i) L213-215 & Fig. 5: Contours and their units are not explained in the caption. Shades of blue in the contours appear to represent ascending motions and have negative values as expected (units Pa/s). But, unless I miss something, anomalies are expressed in units (-Pa/s), which is confusing.

**Reply**: This is the (new) Figure 6 now. The old units where Pa/s. We changed the units to - Pa/s, and for this reason the coloring is reversed now. We have also added the climatology for 20CR. We have also changed (new) Figure 7 to comply with the new units.

ii) L215-217: The monsoon desert mechanism is not thought (by the studies mentioned here) to represent a closed circulation or overturning circulation (Walker-type circulation) with ascending motions over south Asia and descending motions over the EMed. If anything, Rodwell and Hoskins (1996) and Tyrlis et al (2013) present evidence corroborating the notion that the monsoon induces a zonal asymmetry that interacts with the mid- latitude westerlies resulting in enhanced subsidence over the EMed. See for example the discussion in pgs 1396-1397 in Rodwell and Hoskins (1996).

**Reply**. Thank you for this comment. We have deleted the part of the sentence about the closed circulation.

iii) L223-224: "as perhaps expected from the strongest decline in the Etesian winds". This inference is not clear to me. How is the decline in the Etesians related to the anomalies in ascent and descent over India and the EMed?

**Reply**: The descending in the EMed and the adiabatic heating is linked to Etesian winds which advent cool air as demonstrated by Tyrlis and Lelieveld (2013). So it's is likely that a strong anomaly in the descending could be associated with a strong reduction in Etesians. To simplify the manuscript, we deleted this sentence.

iv) L229-230: Actually there is no clear reduction of ascending motions over the "box region" in Fig. 5e. I can see a blue area over continental India & Bay of Bengal and a red area further to the south.

**Reply**: The (new) Figure 6 has been recolored.

v) L237-241 & Fig. 6: Please consider carefully the description of units. In the caption units are referred to as (- Pa/s) while in the label of the horizontal axis as (Pa/s).

**Reply**: Related to a previous comment. Units have been changed to -Pa/s.

vi) L234-234: "Moreover, we identify an almost linear relationship between changes in ISM strength and NED anomalies". I am looking for a punch line in this paragraph describing the results shown in Fig. 6. Is it that there is a linear relationship between ISM strength and NED anomalies or that stronger volcanic eruptions can produce a stronger decline in the frequency Etesians? To put in another way, Fig. 6 is composed of years when volcanic eruptions occurred and it does not describe the climatological strength of the monsoon-desert mechanism, as inferred by the above sentence.

**Reply**: We have changed the units in ISM index to -Pa/s. We have also changed the text and extended the discussion.

7) L245-247: Here the authors infer that cooling over the EMed is due to reduced adiabatic heating. Although that someone would expect that a reduction in the subsidence over the

region would lead to a reduction of the adiabatic warming, evidence about this is not provided in any of the figures. Interestingly, in the Introduction (L33-34), it is mentioned that the Etesians have a cooling effect over the EMed and someone could expect that their decline would be associated with surface warming and not surface cooling, as depicted in Fig. 4a-c. More detailed analysis is required before inferences, such as the above, are reached.

**Reply**: This is true. We don't give any metric for the heating because this would require an additional decomposition of the effects as in the work of Tyrlis and Lelieveld (2013), which is beyond the scope of this study. However, based on earlier work we provide a physical reasoning that explains the detected changes. Its correct that a decline of the Etesians should cause a warming and perhaps this explains the reduced cooling in the North Africa. However, we speculate that this effect is compensated by a direct volcanic cooling and a dynamical cooling from the reduced downwelling. The latter likely explains the amplified cooling in the EMed over land (Sup. Figure 3)

8) L251-253: I do not agree that volcanic eruptions have a trivial effect on SLP over the Balkans. Figs 4a-c suggest that negative SLP anomalies prevail over Europe and expand towards the Balkans. This anomaly contributes to the weakening of the pressure gradient over the EMed. Thus, the decline of the Etesians is not caused only because the thermal low over the Middle East becomes weaker.

9) L18-20, L72 & L279-280: Could the authors comment on the dynamics that cause the weakening of the "Anatolian low"? As mentioned above, it is evident from Figs 4a-c that actually the impact of the volcanic eruption is the appearance of a dipole of SLP, with negative SLP anomaly over Europe and positive SLP over the Middle East.

**Reply**: The last two comments are related. As we commented before, simulations show stronger SLP anomalies over the EMed than Balkan. Perhaps the positive SLP anomalies which seem to extend from the North Atlantic is related to the summertime NAO, but we think this mechanism might play some role in very strong eruptions, such as Samalas. In contrast, the SLP anomalies over the Central Europe and Balkans are trivial in the case of Tambora, and this leads us to argue about the importance of the positive SLP response over the EMed and Anatolian low

**Minor comments**

L12-14: This is a long sentence and a bit difficult to understand, please rewrite.

**Reply**: Abstract has been changed.

L18: Late summer months?

**Reply**: Perhaps it is confusing. It is changed to "We provide model evidence for significant volcanic signatures, manifested as a robust reduction of the average wind speed and the total number of days with Etesian winds in July and August."

L18-20: This is a long and complicated sentence. As mentioned above, I find it difficult to understand the dynamical link between the weakening of the monsoon-desert mechanism and the weakening of the Anatolian low.

**Reply**: Changed

L26: What does "Balkan high" refer to?

**Reply**: It is changed to "*between the central Europe and the Balkans high pressure and the Anatolian low pressure systems*"

L27-29: Something is missing in this sentence. Please rephrase.

**Reply**: Changed

L30: What do the authors mean by "as they are synchronized with the summer monsoon"? With the Indian summer monsoon?

**Reply**: We added the "Indian"

L30-31: What do the authors mean here by "synoptic system"?

**Reply**: Is related to the extent of the ' surface pressure gradient' in EMed

L32-33: It is not clear how the Etesians are "amplified by a large-scale subsidence established in summer months under the influence of the Indian and Asian summer monsoon". Please clarify.

**Reply**: Changed to "*This synoptic system and particularly the Anatolian low pressure system is frequently viewed as the westernmost extension of the Persian trough (e.g. Bollasina and Nigam 2011). Etesians advect cool air masses to the Aegean Sea and Levant to…*"

L37: "increased atmospheric blocking activity over Europe"

**Reply:** Changed

L42-43: What do the authors imply by "vice versa"? I think that this sentence could be removed.

**Reply**: 'vice versa' is removed

L75-77: This is a confusing sentence. Please rewrite.

**Reply**: changed to "We conclude by discussing the implications of our results for near-term prediction and decadal Etesian wind changes in a warming climate. "

L91: Does "1" correspond to a footnote?

**Reply**: yes. There was a footnote in Page 3. Now it is moved in the main text in parenthesis.

L119-122: Long and confusing sentence. Please rewrite.

**Reply**: The paragraph is re-written to include discussion for the (new) Figure 1.

L131: What is the meaning of "muted" and "punctuated" here?

**Reply**: Changed to "Periods of weak interannual variability are interrupted by periods…"

L155: "In addition" - > "For completeness, "

**Reply**: Changed

L230: "in negligible" - > "is negligible"

**Reply**: Changed

L259: Do you mean here "anomalous temperature gradients"? Please clarify.

Reply: changed to "*cause strong equator-to-pole gradients of temperature anomalies*"

L261-262: I might be wrong. But isn't Pinatubo a volcano in the tropical region?

**Reply**: True. We meant to say a NH eruption of Pinatubo magnitude. Changed to "NH eruption of Pinatubo magnitude might cause a considerable reduction in Etesian winds owing to the amplified hemispheric sensitivity."

L271: "positive?"?

**Reply**: ? is deleted

Caption of Fig. 1: "volcaninc"->"volcanic"

**Reply**: Changed

Caption of Fig. 2: Please add explanations for orange and red lines.

Reply: changed.

L12, 19, 23, 25, 44, 56, 73, 120 and possibly elsewhere: It may be correct to write "over the eastern Mediterranean" or "over the central Aegean".

L12 and elsewhere: It may be better to replace "Etesian winds" with "Etesians".

**Reply**: Changed through the text except the abstract

---

## Author Response (AR2)

**Editor:**

- acknowledgements: it feels appropriate to thank the reviewers for their very detailed and constructive comments that helped greatly to improve the paper. **Reply:** Yes indeed. Added

- line 15: "The ... winds is ..." is grammatically incorrect, either "The winds are" or "The wind is". Please check the entire paper for similar grammatical issues! **Reply:** changed

- line 15: "emerging from" **Reply:** changed

- line 22: maybe add year for the Samalas eruption **Reply:** added

- line 261: "Large eruptions are ideal ..." **Reply:** changed

- line 268: "that reduce adiabatic heating" **Reply:** changed

- line 271: "the ensemble averaging ... suppresses ..." **Reply:** changed

- line 275: I don't understand "provided the internal variability had been negligible". If your simulation did not show Etesians in this year, then this part of the sentence can be deleted(?). **Reply:** the modified sentence now reads:

These circulation changes are clearly detected after the largest eruptions in the past millennium. According to the CESM-LME, the first post-eruption year after Samalas in the observation record should have been a summer without Etesians, provided the internal variability had been negligible.

- line 279 and in other places: "SLPs" does not make sense, there is no plural of "sea level pressure" **Reply:** changed

**Keply:** changed

- line 282: do you mean "cooling in the tropics"? **Reply: changed**

- line 291: do you mean "for some of the unknown eruptions"? **Reply: changed**

- caption of Fig. 1: write "m/s" as "m s^{-1}" **Reply:** Not sure why should be written in this way. Is it to follow the journal's style? Reviewer 1

Specific Comments L22: "in the extreme case of the eruption of the Samalas" **Reply:** Kept original

L31-32: "between the high-pressure systems covering central Europe and the Balkans and the Anatolian low-pressure system" **Reply:** changed

L33: "the topography of the EMed channels"? **Reply:** Has the meaning of directing the winds

L34: "northwesterly direction" Introduction and elsewhere: Please replace "Etesians" by "The Etesians" **Reply:** changed

L39: "over the Aegean Sea" **Reply: c**hanged

L41,48, 49 and elsewhere: I am not a native speaker of English myself but I think that since the focus of the study is on atmospheric responses (and not in the ocean), it is more appropriate to write "over the EMed". The authors may wish to check with a native speaker and change this throughout the manuscript. **Reply:** changed

L47: "an important source that modulates interannual variability"? **Reply:** it now reads as *as an important component of Etesian winds variability on interannual time scales*

L57: "Chronis et al. (2011).... You may want to start a new sentence here. **Reply:** kept original

L79-80: This is a bit awkward sentence. Please rephrase.

Reply: it now reads as

we present model evidence for a significant decline of the Etesian winds in response to volcanic eruptions over the last millennium, with a stronger sensitivity to NH eruptions

L103-107: This is a very long sentence with many parentheses that makes it difficult to understand. Please rephrase. **Reply:** Moved parentheses to the end of the sentence.

L127: "Etesian winds" **Reply:** Not sure what is commented here

L145: "We first analyse.." **Reply:** changed

L147-150: Please rephrase this long sentence. **Reply:** not changed

Figure 5 and Section 3.2: I might miss something but the anomalies following the eruption of Tambora (Fig. 5c) receive minimum attention in this section. However, the anomalies both over the eastern Mediterranean and India are statistically significant. Please comment. **Reply:** Yes we agree with the reviewer that responses are significant for Tambora as well as for other eruptions presented as shown in the supplementary Figures. We perhaps discussed more extensively effects after the Laki eruption because results are more surprising given its considerably weaker magnitude compared to Tambora. This demonstrates the amplified sensitivity to NH eruptions.

L217-219: Causality is not addressed by this study so it may be better to write "This indicates a weakened SLP pressure gradient over the Aegean Sea, which is associated with reduced wind speeds, as evidenced with the southerly anomalies of about 1 m/s (arrows in Figure 5 and Sup. Figure 2). **Reply:** changed

L223: "increasing shortwave heating"? **Reply:** Changed to *This can also explain the anomalous surface warming simulated over India given that a reduction in the cloud amount and increased downward shortwave radiation in years of reduced ISM can cause positive surface temperature anomalies.*

L225-227: I am sorry but I still cannot figure out where to look for the anomalous northwesterly anomalies over the Arabian Sea in Fig. 5e. Is it near the coast of Saudi Arabia? Please clarify. **Reply:** changed

L247: "Figure 6e" **Reply:** changed

L282: "given that they induce negligible cooling to the tropics" **Reply:** changed

Fig 1e, Fig. 4 & L471: I would recommend the use units of "hPa" throughout the manuscript. **Reply:** We kept the original units in Pa.